# Spatiotemporal variability in pH and carbonate parameters on the Canadian Atlantic continental shelf between 2014 and 2022

**Olivia Gibb[1], Frédéric Cyr[1], Kumiko Azetsu-Scott[2], Joël Chassé[3], Darlene Childs[2], Carrie-Ellen Gabriel[2], Peter S. Galbraith[4], Gary Maillet[1], Pierre Pepin[1], Stephen Punshon[2], and Michel Starr[4]**

[1]Northwest Atlantic Fisheries Centre, Fisheries and Oceans Canada, St. John's, NL, Canada
[2]Bedford Institute of Oceanography, Fisheries and Oceans Canada, Dartmouth, NS, Canada
[3]Gulf Fisheries Centre, Fisheries and Oceans Canada, Moncton, NB, Canada
[4]Maurice Lamontagne Institute, Fisheries and Oceans Canada, Mont-Joli, QC, Canada

**Correspondence:** Frédéric Cyr (frederic.cyr@dfo-mpo.gc.ca)

**Abstract.** The Atlantic Zone Monitoring Program (AZMP) was established by Fisheries and Oceans Canada (DFO) in 1998 with the aim of monitoring physical and biological ocean conditions in Atlantic Canada in support of fisheries management. Since 2014, at least two of the carbonate parameters (pH; total alkalinity, TA; and dissolved inorganic carbon, DIC) have also been systematically measured as part of the AZMP, enabling the calculation of derived parameters (e.g., carbonate saturation states, $\Omega$, and partial pressure of $CO_2$, $p\mathrm{CO_2}$). The present study gives an overview of the spatiotemporal variability in these parameters between 2014 and 2022. Results show that the variability in the carbonate system reflects changes in both physical (e.g., temperature and salinity) and biological (e.g., plankton photosynthesis and respiration) parameters. For example, most of the region undergoes a seasonal warming and freshening. While the former will tend to increase $\Omega$, the latter will decrease both TA and $\Omega$. Spring and summer plankton blooms decrease DIC near the surface and then remineralize and increase DIC at depth in the fall. The lowest $p\mathrm{CO_2}$ values (down to $\sim 200\,\mu\mathrm{atm}$) are located in the cold coastal Labrador Current, whereas the highest values ($> 1500\,\mu\mathrm{atm}$) are found in the fresh waters of the Gulf of St. Lawrence and the St. Lawrence Estuary. The latter is also host to the lowest pH values of the zone (7.48 in the fall of 2022). Finally, most of the bottom waters of the Gulf of St. Lawrence ($> 90\,\%$) are undersaturated with respect to aragonite ($\Omega_{\mathrm{arg}} < 1$). In addition to providing a baseline of carbonate parameters for the Atlantic Zone as a whole, this comprehensive overview is a necessary and useful contribution for the modelling community and for more in-depth studies. The full dataset of measured and derived parameters is available from the Federated Research Data Repository: https://doi.org/10.20383/102.0673 (Cyr et al., 2022a).

## 1 Introduction

The Canadian Atlantic continental shelf and slope, termed the Canadian Atlantic Zone (or simply the Atlantic Zone hereafter), is at the confluence of waters of contrasting origin (Fig. 1). The northward-flowing warm and saline Gulf Stream, the southward-flowing cold and less saline Labrador Current, and the fresh outflow from the St. Lawrence River and the Gulf of St. Lawrence provide a unique range of salinities and temperatures (Fig. 2a) that form oceanic fronts along the shelf–slope edge surrounding the Atlantic Zone (Belkin et al., 2009; Cyr and Larouche, 2015). These fronts are often associated with vertical nutrient transport, high productivity and ecological hot spots with a rich diversity and abundance of organisms (e.g., Lévy et al., 2012). As a result, the Atlantic Zone supports a variety of commercially impor-

tant species which amounted to nearly CAD 5 billion in commercial landings (CAD 4.2 billion) and finfish and shellfish aquaculture (CAD 0.6 billion) in 2021 (DFO, 2022). However, these ecosystems are becoming stressed due to changes in the physical, chemical and biological oceanographic environment as a result of industrialization, e.g., the burning of fossil fuels, agriculture and deforestation (Bernier et al., 2018; Doney, 2010). Increasing atmospheric carbon dioxide ($CO_2$) concentrations not only increase atmospheric and sea surface temperatures but also alter the ocean's carbonate cycle, thereby resulting in increased ocean acidity (i.e., decreased pH). Many aquatic species within the Atlantic Zone are vulnerable to the effects of ocean acidification, which may affect the entire ecosystem (Fabry et al., 2008; Doney, 2010). Calcifying shellfish are directly impacted by ocean acidification, whereas other shellfish and finfish can be indirectly impacted (Kroeker et al., 2013). The socioeconomic impacts of ocean acidification and climate change on fisheries (e.g., resource decline, changes in species distribution, and income and cultural losses) have been assessed for various ecosystems (Cooley and Doney, 2009; Mathis et al., 2015; Wilson et al., 2020). Monitoring the physical, chemical and biological oceanographic conditions and determining a baseline is required prior to an assessment of future aquaculture and fisheries in the Atlantic Zone.

Implemented in 1998 by Fisheries and Oceans Canada (DFO), the Atlantic Zone Monitoring Program (AZMP) was designed to evaluate the physical (temperature and salinity), chemical (nutrients and dissolved oxygen concentrations) and biological (chlorophyll-*a*, fluorescence, and plankton species assemblage and abundance) oceanographic properties of the Canadian northwestern Atlantic (Therriault et al., 1998). The AZMP characterizes the spatial and temporal (seasonal to decadal) variability in these oceanic properties. It is carried out by the four Atlantic DFO administration centres (three geographic regions): Québec/Gulf, Maritimes, and Newfoundland and Labrador (Fig. 3). The AZMP annually publishes a summary of oceanographic conditions that assesses the current state of the ecosystem. It is used for stock assessments and marine resource management by DFO and also supports fundamental oceanographic research. In fall of 2014, sampling and analysis of two of the carbonate parameters, including total alkalinity (TA), total dissolved inorganic carbon (DIC) and pH, were added to the AZMP mandate. The resulting dataset is the focus of the present paper.

Ocean acidification is the decrease in ocean pH (increase in $H^+$, acidity) and carbonate ion concentration ($CO_3^{2-}$) due to the increased uptake of anthropogenic $CO_2$ (Millero, 2007). Seawater carbonate chemistry involves the dissolution of atmospheric $CO_2$ (Eq. 1), acid–base reactions forming the inorganic carbonate (DIC) species in equation (Eq. 2), and the formation and dissolution of solid calcium carbonate ($CaCO_3$) (Eq. 3).

$$CO_{2_{atm}} \rightleftharpoons CO_{2_{aq}} \tag{1}$$

$$CO_{2_{aq}} + H_2O \rightleftharpoons H^+ + HCO_3^-$$

$$HCO_3^- \rightleftharpoons H^+ + CO_3^{2-} \tag{2}$$

$$CaCO_3 \rightleftharpoons \left[Ca^{2+}\right] + \left[CO_3^{2-}\right] \tag{3}$$

The photosynthesis and respiration of organic matter is another factor that contributes to changes in the carbonate system (Eq. 4). Spring phytoplankton blooms increase surface water pH by consuming $CO_2$ in water, while the remineralization of that vertically exported organic matter to DIC lowers pH at depth.

$$CH_2O + O_2 \rightleftharpoons CO_2 + H_2O \tag{4}$$

The degree to which seawater is saturated with calcium carbonate ($CaCO_3$) is known as the $CaCO_3$ saturation state ($\Omega$). The saturation state is a function of calcium and carbonate concentrations, and its apparent solubility product ($K_{sp}^*$) which is a function of temperature, salinity and pressure, is expressed as follows (Mucci, 1983; Millero, 1995):

$$\Omega = \frac{\left[Ca^{2+}\right]\left[CO_3^{2-}\right]}{K_{sp}^*}. \tag{5}$$

Seawater with an $\Omega > 1$ is considered oversaturated, inducing carbonate precipitation, whereas water with $\Omega < 1$ is considered undersaturated and corrosive, promoting carbonate dissolution. The carbonate saturation state customarily decreases with depth as a result of cold temperatures, increased pressure and respired DIC. The depth at which the water becomes undersaturated is generally referred to as the *saturation horizon*. Organisms such as phytoplankton, zooplankton and invertebrates (e.g., clams, oysters and corals) that form shells and skeletons of calcium carbonate ($CaCO_3$) may have difficulty maintaining or forming hard structures in undersaturated waters. It should be noted, however, that several studies have identified higher critical thresholds ($\Omega \sim$ 1.3–2) for marine organisms (e.g., Ekstrom et al., 2015; Waldbusser et al., 2015; Siedlecki et al., 2021). The two most common forms of $CaCO_3$ are calcite and aragonite. Because aragonite is more soluble than calcite, the aragonite saturation horizon ($\Omega = 1$) is shallower, and organisms that produce aragonite may be more vulnerable to a decreasing saturation state as atmospheric $CO_2$ increases.

Freshwater influx also plays an important role in the carbonate system. The effect of freshwater on $\Omega$ is due to both changes in the DIC/TA ratio in source waters and the decrease in the $Ca^+$ concentration as a function of salinity (e.g., Azetsu-Scott et al., 2014; Hunt et al., 2021). TA is an indicator of seawater's ability to buffer (neutralize) acids. It is a measure of the excess total proton acceptors (anions) over proton donors (acids) formed by the dissociation of carbonic,

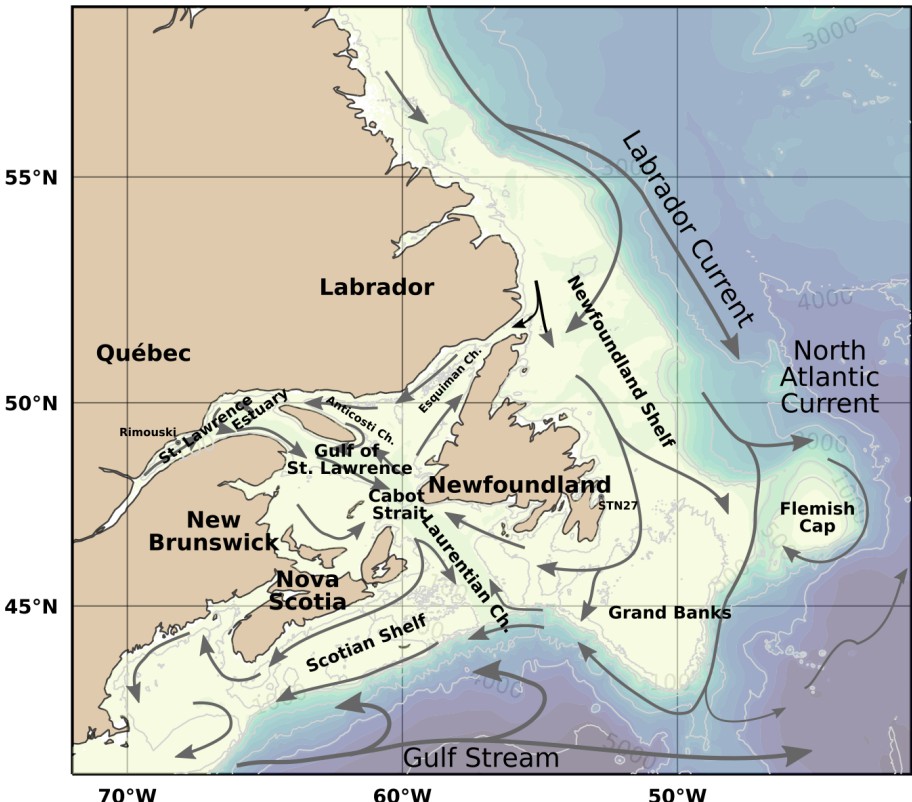

**Figure 1.** Bathymetric map of the Atlantic Zone (i.e., the Newfoundland and Labrador shelves and slopes, the Scotian Shelf and Slope, the Gulf of St. Lawrence, and the St. Lawrence Estuary) CE1 . Also identified are the deep channels within the Gulf of St. Lawrence: the Laurentian, Anticosti and Esquiman channels. The approximate position and direction of the major current systems and their shelf and slope components are illustrated as arrows: the Gulf Stream (grey) and the Labrador Current (black).

boric and other weak acids in seawater. It is expressed as mole equivalents of hydrogen ions per kilogram of seawater. Therefore, TA will decrease with freshwater influx and during carbonate precipitation (decrease in DIC), which will subsequently decrease $\Omega$:

$$CO_2 + CaCO_3 + H_2O \rightleftharpoons 2HCO_3^- + Ca^{2+}. \qquad (6)$$

## 2 Physical oceanographic setting

The physical properties of the Atlantic Zone are dominated by large-scale interactions of Arctic, subpolar and subtropical waters and freshwater influx from the St. Lawrence River (e.g., Loder et al., 1998; Han et al., 2008; Brickman et al., 2016). The geographic and current names described below are shown in Fig. 1, and the different water masses (or water origins) of this system are loosely highlighted in a temperature vs. salinity ($T-S$) and depth plot of the entire dataset (Fig. 2a). The two major circulation features are the southward-flowing Labrador Current, which brings cold and less saline subpolar waters along the Newfoundland and Labrador (NL) Shelf, and the northeastward-flowing Gulf Stream (with the North Atlantic Current, NAC, as its north-

east extension), which carries warm and saline subtropical waters along the Scotian Shelf and the Grand Banks. The latter is labelled North Atlantic Water (NAtlW) in Fig. 2. An additional input of Arctic-origin cooler and fresher water also flows on the inshore part of the Newfoundland and Labrador Shelf, a current generally referred to as the inshore Labrador Current or the coastal Labrador Current (Florindo-López et al., 2020). These contrasting water masses interact on the shelf and partially enter the Gulf of St. Lawrence (GSL), a semi-enclosed basin characterized by an estuarine circulation.

On the Scotian Shelf, Slope Water that lies along the shelf break is a combination of cold, less saline, oxygen-rich and nutrient-poor Labrador Slope Water (LSW) from the Labrador Current and warm and saline, nutrient-rich and oxygen-poor Warm Slope Water (WSW) derived from the NAC. The properties of the Slope Water vary depending on which component (LSW vs. WSW) is dominant (Petrie and Drinkwater, 1993). This mixture is advected inland in the deep waters of the Laurentian Channel by the estuarine circulation of the GSL. As these waters flow landward toward the Lower St. Lawrence Estuary (LSLE), oxygen is consumed and the deep water becomes hypoxic (Gilbert et al., 2005).

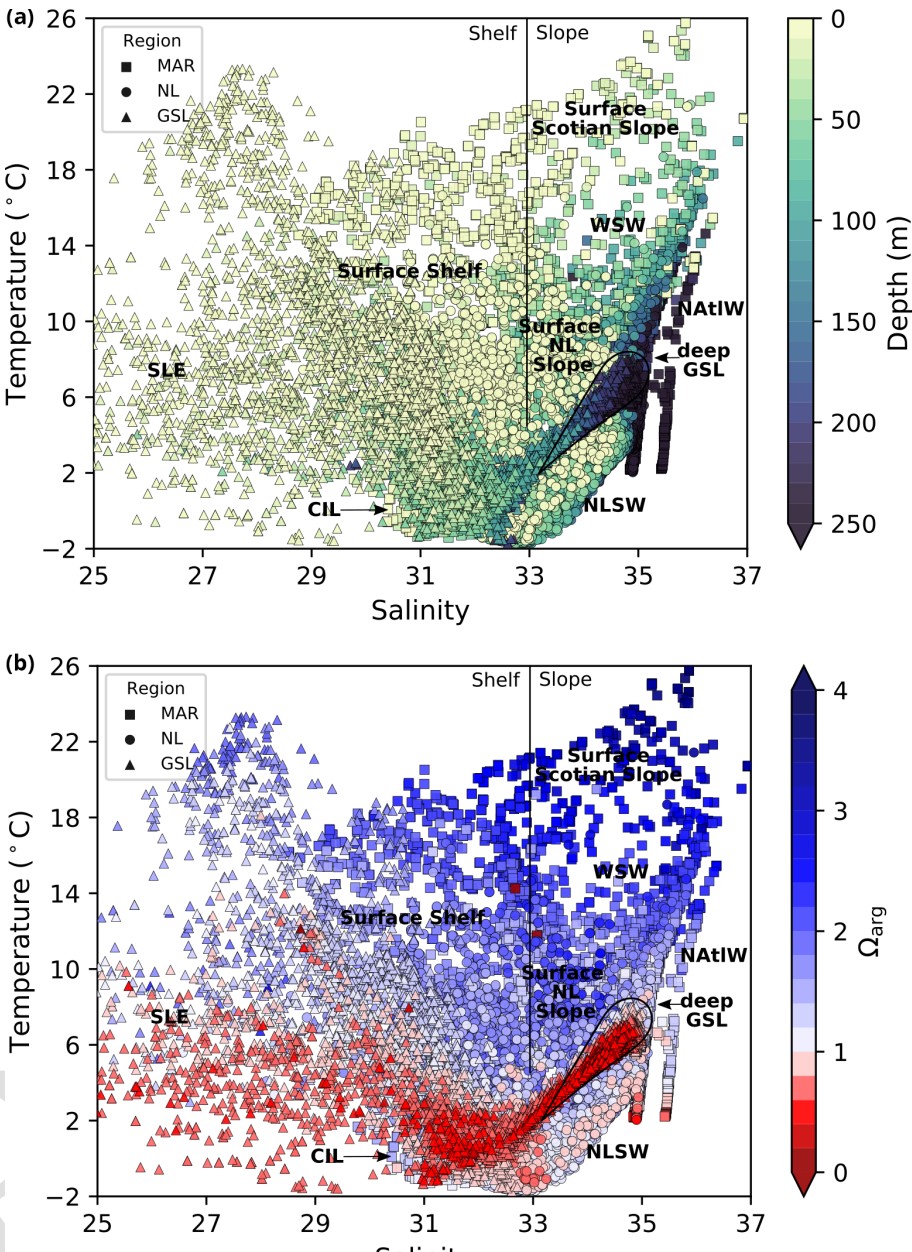

**Figure 2.** Diagrams of temperature and salinity vs. depth **(a)** and temperature and salinity vs. saturation state relative to aragonite for all bottle samples in the dataset. The three regions of the Atlantic Zone are shown using different markers. An approximate division between the shelf and slope surface waters is indicated by the vertical black line at a salinity value of 33. Various water masses are labelled as follows: WSW, Warm Slope Water; NAtlW, North Atlantic Water; and NLSW, Newfoundland and Labrador Slope Water. In the absence of a clear water mass definition, the origin of the water is labelled as follows: LSLE, Lower St. Lawrence Estuary; GSL, Gulf of St. Lawrence; and LS, Labrador Shelf.

These low oxygen concentrations have been associated with an increased composition ratio of WSW relative to LSW (Gilbert et al., 2005; Jutras et al., 2020), poor ventilation (> 16 years; Mucci et al., 2011) and increased oxygen demand in bottom waters because of respiration and remineralization of organic matter supplied by increased primary productivity at the surface (Thibodeau et al., 2006). The Scotian Shelf Water receives subpolar water primarily from the GSL through the western Cabot Strait as well as a smaller portion directly from the Newfoundland Shelf by the Labrador Current across the Laurentian Channel (Dever et al., 2016).

In the summer, the water column of the Atlantic Zone is generally characterized by a three-layer system. A seasonally warmed thin and relatively fresh surface layer overlies

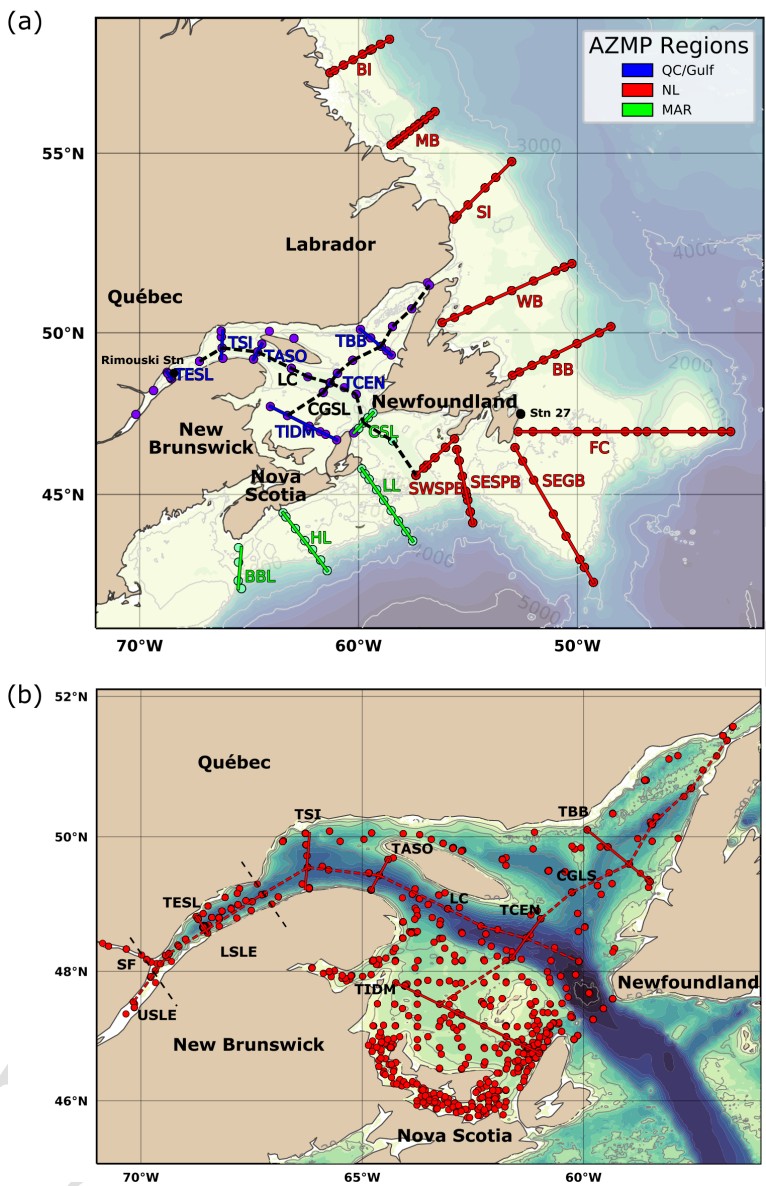

**Figure 3.** Bathymetric maps of the Atlantic Zone. AZMP surveys of the four regions, including Maritimes (MAR), Newfoundland and Labrador (NL), and Gulf of St. Lawrence (GSL). The map of the Atlantic Zone in panel **(a)** shows the locations of the following standard AZMP sections, from north to south and from west to east across the entire GSL (e.g., Fig. B5): Beachy Island (BI), Makkovik Bank (MB), Seal Island (SI), White Bay (WB), Bonavista Bay (BB), Flemish Cap (FC), Southeast Grand Bank (SEGB), Southeast St. Pierre Bank (SESPB), Southwest St. Pierre Bank (SWSPB), Cabot Strait Line (CSL), Louisbourg Line (LL), Halifax Line (HL), Browns Bank Line (BBL), Transect Bonne Bay (TBB), Transect Central (TCEN), Transect Îles de la Madeleine (TIDM), Transect Anticosti South (TASO), Transect Sept-Îles (TSI), Transect Estuaire du Saint-Laurent (TESL) and the Laurentian Channel (LC). Panel **(b)** zooms in on the GSL region and includes other sampling locations from the dataset, such as the summer groundfish surveys. Here, the Lower St. Lawrence Estuary (LSLE) corresponds to the upstream portion of the GSL. The Upper St. Lawrence Estuary (USLE) and the Saguenay Fjord (SF) are also annotated.

a cold intermediate layer (CIL), which is the remnant of the previous winter's cold surface layer. A warmer, more saline, slope-derived water lies below the CIL. The deeper waters (> 200 m) are generally limited to channels and troughs, and there are areas (e.g., southwestern Scotian Shelf) where the CIL is absent due to mixing. In the late fall and winter, the surface water cools, deepens and combines with the CIL into a two-layer system. In places, this winter mixed layer reaches the freezing point and sea ice is formed. This winter layer is at the origin of the following summer CIL (Galbraith, 2006).

Thus, the Atlantic Zone is characterized by a large range of temperatures (from the freezing point to over 25 °C during the summer) and salinities (from freshwater to a salinity value of over 37); this influences the carbonate system, as discussed later. The Atlantic Zone is geographically divided into three regions here, and it is then further divided using bathymetrical considerations: shelf (fresher) vs. slope (more saline) and surface (warm in summer) vs. subsurface (cool).

## 3    Sampling and methodology

The dataset presented here consists of carbonate parameters compiled from multiple surveys from all four administrative AZMP regions: Newfoundland and Labrador (NL), Maritimes (MAR), Québec (QC) and Gulf. As both the QC and Gulf regions sample the Gulf of St. Lawrence, they will be combined under the name GSL herein (e.g., Fig. 3). The map of the Atlantic Zone in Fig. 3a shows the locations of the standard AZMP sections as well as two sections through the GSL established here to plot depth profiles across (GSL; from the Magdalen Shallows in the southwest to the Strait of Belle Isle in the northeast) and along the Laurentian Channel (LC; the deep channel that runs from Cabot Strait to the head of the LSLE), shown magnified in Fig. 3b. For the purpose of presenting the data, the Cabot Strait Line (CSL) will be associated with the GSL. Unless explicitly stated, the waters of the Upper St. Lawrence Estuary (USLE) and the Saguenay Fjord (SF; Fig. 3b) are not discussed in this study, as their properties, which are highly influenced by freshwater input, are very different from the rest of the Atlantic Zone. These data are, however, included in the dataset that accompanies this paper. All bathymetric maps were generated using the GEBCO 2014 Grid bathymetry product (version 20150318, http://www.gebco.net, last access: 10 April 2019).

### 3.1   Sampling

Carbonate chemistry data have been sampled in the Atlantic Zone since the fall of 2014 (except for the high-frequency Rimouski station where the sampling started earlier; see below). Depending on the region, between two and three surveys have usually occurred every year since. An overview of all missions, their dates and the parameters sampled is provided in Table 1.

The NL region, due to its extensive shelf and slope spatial extent, has a survey schedule that surveys the NL region three times a year, generally from the Southwest St. Pierre Bank (SWSPB) section up to the northern sea ice extent: usually the Bonavista Bay (BB) section during spring (April) surveys, from the Flemish Cap (FC) section to Beachy Island (BI) during summer (July) surveys and from the SWSPB section to Seal Island (SI) during fall (November) surveys (Fig. 3a). The surveys are most often incomplete due to operational constraints caused by sea ice, weather or research vessel downtime. The NL region survey schedule aims to

sample at the surface (∼ 5 m), 10, 50, 75, 100, 150, 250, 500, 1000 m and the bottom. Sampling down the shelf slope to depths greater than 1200 m was, however, only feasible during fall surveys (maximum 3677 m) because of limited vessel capacity during other seasons.

The Maritimes region has respective spring (April) and fall (September/October, except November/December in 2017) surveys, which have been consistently carried out between 2014 and 2022 (Fig. 3a). The Scotian Shelf and Slope were sampled at the surface (∼ 2 m), 10, 50, 80, 100, 150, 250, 500, 1000, 1500 m and the bottom.

The GSL is sampled during spring (June) and fall (October/November) surveys along the standard sections at the surface (∼ 2 m), 15, 50, 100, 150, 200, 300, 400 m and the bottom (Fig. 3b). Samples have also been acquired during groundfish surveys in the GSL (Fig. 3b, Table 1) during summer (August/September) 2017–2022 (Trip_Name: "IML2017027", "PER2017018", "PER2017024", "PER2017016", "PER2017026", "MLB2017001b", "IML201841", "PER2018041", "PER2018042", "PER2018043", "TEL2018196", "IML2019036", "PER2019140", "PER2019008", "TEL2019201", "PER2020019", "PER2021156", "PER2021151", "CJC2021222", "IML2021030", "PER2021021", "IML2022039", "PER2022022", "CAR2022025" and "PER202223330"). Although these surveys are not within the regular AZMP program, they are staffed by the AZMP for oceanographic sampling and provide extensive spatial and added temporal (summer) coverage of carbonate parameters for the GSL.

The fall 2014 and spring 2017 AZMP surveys had the most complete spatial coverage of the Atlantic Zone among the entire time series. The other surveys are partial due to operational limitations specific to each region (see Table 1 for the timing of the different surveys per region).

In addition, two coastal stations are sampled at a greater temporal frequency. Rimouski station (labelled "Rimouski" in Figs. 1 and 3), located in the LSLE, is sampled by the Québec region on a near-weekly basis. Station 27 (labelled "STN27" in Figs. 1 and 3), located on the path of the coastal Labrador Current just outside St. John's Harbour is sampled by the NL region on a near-monthly basis. To illustrate their respective temporal coverage, the time series of pH at these stations (measured at Rimouski station and derived from TA–DIC[CE3] at Station 27) are presented in Fig. 4. Note that Rimouski station is the only time series starting in the spring of 2014 in this dataset (all other sampling starts in the fall of 2014).

The collection of discrete water samples for carbonate analyses was achieved using a 12 or 24 Niskin bottle rosette equipped with a SeaBird conductivity–temperature–depth (CTD) unit coupled with an oxygen sensor to continuously measure depth, temperature, salinity and the dissolved oxygen (DO) concentration. Aliquots of water were subsampled from each Niskin bottle for nutrients ($PO_4$, $SiO_3$ and

**Table 1.** Summary of all Atlantic Zone surveys included in this dataset that collected carbonate chemistry data from fall 2014 to 2022. Note that the number of bottles corresponds to samples that include at least one of the carbonate parameters (TA, DIC or pH) and might not signify that the full suite of parameters was derived using CO2SYS. Targeted measured parameters for each mission are provided in the last column. Please note that all dates in the table are given in the following format: yyyy-mm-dd. CE2

| Year | Season | Region | Trip name | Stations | Bottles | Max depth (m) | | End date | Measured parameters |
|------|--------|--------|-----------|----------|---------|---------------|------|----------|---------------------|
| 2014 | Spring | GSL | M14010 | 1 | 43 | 330 | 2014-04-25 | 2014-12-16 | TA, pH |
| | Fall | GSL | IML201437 | 57 | 316 | 465 | 2014-10-27 | 2014-11-09 | TA, DIC, pH |
| | | MAR | 18HU14030 | 37 | 241 | 2905 | 2014-09-19 | 2014-10-08 | TA, DIC, pH |
| | | NL | HUD114 | 37 | 254 | 1071 | 2014-11-16 | 2014-12-07 | TA, DIC, pH |
| 2015 | Spring | GSL | IML2015010 | 1 | 174 | 331 | 2015-01-18 | 2015-12-18 | TA, pH |
| | | | IML201516 | 9 | 51 | 452 | 2015-06-01 | 2015-06-12 | pH |
| | | MAR | 18HU15004 | 21 | 142 | 2941 | 2015-04-18 | 2015-04-25 | TA, DIC |
| | | NL | TEL144 | 26 | 137 | 533 | 2015-04-12 | 2015-04-27 | TA, DIC |
| | Summer | NL | TEL148 | 43 | 255 | 1009 | 2015-07-09 | 2015-07-26 | TA, DIC |
| | Fall | GSL | IML2015041 | 14 | 85 | 464 | 2015-10-19 | 2015-11-05 | TA, pH |
| | | MAR | 18HU15030 | 51 | 386 | 5044 | 2015-09-20 | 2015-10-08 | TA, DIC |
| | | NL | HUD115 | 43 | 254 | 3676 | 2015-11-15 | 2015-12-06 | TA, DIC |
| 2016 | Spring | GSL | IML2016010 | 1 | 201 | 333 | 2016-01-23 | 2016-12-12 | TA, pH |
| | | | IML2016015 | 19 | 110 | 459 | 2016-06-01 | 2016-06-26 | TA, pH |
| | | MAR | 18HU16003 | 15 | 106 | 1254 | 2016-04-10 | 2016-04-25 | TA, DIC |
| | | | 18HU16006 | 15 | 191 | 4717 | 2016-05-02 | 2016-05-24 | TA, DIC |
| | | NL | TEL157 | 7 | 32 | 446 | 2016-04-01 | 2016-04-06 | TA, DIC |
| | | | TEL159 | 15 | 104 | 1210 | 2016-05-11 | 2016-05-17 | TA, DIC |
| | Summer | NL | TEL160 | 32 | 215 | 1074 | 2016-07-08 | 2016-07-28 | TA, DIC |
| | Fall | GSL | IML2016050 | 55 | 202 | 465 | 2016-10-16 | 2016-11-02 | TA, pH |
| | | MAR | 18HU16027 | 60 | 377 | 3947 | 2016-09-16 | 2016-10-04 | TA, DIC |
| | | NL | HUD116 | 23 | 145 | 3303 | 2016-11-13 | 2016-11-20 | TA, DIC |

$NO_3 + NO_2$) and DO, and they were then processed and analyzed according to established standard protocols (Mitchell et al., 2002). Although the DO concentration (in $mL\,L^{-1}$) is available in this dataset, it was sampled less frequently than the other oceanographic parameters because its purpose was to calibrate the DO sensor measurements. In the GSL region, DO was sampled at the surface and bottom, whereas DO was sampled at the surface, the bottom and middle of the water column in the MAR region. In the NL region, DO was sampled at all depths but at a subsampled number of stations (roughly every two to four stations along a section). In the GSL and NL regions, the subset of bottle DO values was used to calibrate the DO sensor on the CTD unit, providing a DO value for each sample bottle (even when no Winkler titration was realized). For MAR data, only DO data obtained from Winkler titration are provided. Oxygen saturation ($O_{2_{sat}}$, in %) was calculated as the ratio between the measured DO concentration and its solubility referenced to the surface us-

ing the TEOS-10 toolbox (McDougall and Barker, 2011) and corresponding $T–S$ observations.

Aliquots of water for the analysis of pH, total alkalinity (TA) and total dissolved inorganic carbon (DIC) (or a combination thereof) were subsampled following the protocols established by Dickson (2010). Each aliquot was carefully drawn from the Niskin bottles, transferred into 500 mL gas-tight borosilicate reagent bottles (Corning, USA) and allowed to overflow by one volume while precluding air bubbles. A 5 mL volume was then removed to create a headspace for thermal expansion. The samples were preserved with $100\,\mu L$ of saturated $HgCl_2$ to prevent further biological activity. Each bottle was sealed with a ground-glass stopper, high-vacuum (Apiezon M) grease and a rubber band, and they were then stored in the dark between 4 and 18 °C for up to 6 months prior to analysis.

| Year | Season | Region | Trip name | Stations | Bottles | Max depth | Start date | End date | Measured parameters |
|---|---|---|---|---|---|---|---|---|---|
| 2017 | Spring | GSL | IML2017050 | 1 | 170 | 333 | 2017-02-07 | 2017-12-04 | TA, pH |
| | | | IML2017080 | 40 | 224 | 466 | 2017-05-30 | 2017-06-19 | TA, pH |
| | | | PER2017016 | 6 | 18 | 65 | 2017-06-15 | 2017-06-16 | TA, pH |
| | | | PER2017018 | 4 | 8 | 20 | 2017-06-23 | 2017-06-28 | TA, pH |
| | | MAR | 18OL17001 | 46 | 265 | 3000 | 2017-04-19 | 2017-05-01 | TA, DIC |
| | | NL | TEL173 | 51 | 363 | 1208 | 2017-04-06 | 2017-04-23 | TA, DIC |
| | Summer | GSL | PER2017024 | 40 | 80 | 49 | 2017-07-12 | 2017-07-30 | TA, pH |
| | | | IML2017027 | 28 | 148 | 524 | 2017-08-03 | 2017-09-01 | TA, pH |
| | | | PER2017026 | 27 | 81 | 136 | 2017-08-16 | 2017-08-21 | TA, pH |
| | | | MLB2017001B | 9 | 30 | 454 | 2017-08-25 | 2017-08-30 | TA, pH |
| | | NL | TEL176 | 48 | 352 | 1252 | 2017-07-08 | 2017-07-28 | TA, DIC |
| | Fall | GSL | IML2017048 | 46 | 223 | 455 | 2017-11-04 | 2017-11-23 | TA, pH |
| | | MAR | 32EV17606 | 56 | 324 | 3757 | 2017-11-24 | 2017-12-15 | TA, DIC |
| | | NL | DIS009 | 27 | 195 | 1201 | 2017-11-11 | 2017-12-16 | TA, DIC |
| 2018 | Spring | GSL | IML2018040 | 1 | 364 | 331 | 2018-03-12 | 2018-12-06 | TA, pH |
| | | | IML2018014 | 70 | 275 | 458 | 2018-06-05 | 2018-06-24 | TA, pH |
| | | MAR | 18HU18004 | 43 | 320 | 4513 | 2018-04-08 | 2018-04-23 | TA, DIC |
| | | NL | TEL185 | 33 | 213 | 1200 | 2018-04-06 | 2018-04-24 | TA, DIC |
| | Summer | GSL | PER2018041 | 40 | 80 | 45 | 2018-07-11 | 2018-07-30 | TA, pH |
| | | | IML201841 | 36 | 156 | 483 | 2018-08-03 | 2018-08-31 | TA, pH |
| | | | PER2018042 | 32 | 96 | 136 | 2018-08-13 | 2018-08-21 | TA, pH |
| | | NL | COR011 | 80 | 492 | 1202 | 2018-07-15 | 2018-08-02 | TA, DIC |
| | Fall | GSL | TEL2018196 | 36 | 120 | 388 | 2018-09-07 | 2018-10-01 | TA, pH |
| | | | PER2018043 | 6 | 14 | 89 | 2018-09-27 | 2018-10-07 | TA, pH |
| | | | IML2018028 | 31 | 154 | 459 | 2018-10-23 | 2018-11-01 | TA, pH |
| | | MAR | 18HU18030 | 49 | 361 | 4144 | 2018-09-15 | 2018-10-05 | TA, DIC |
| | | NL | HUD118 | 46 | 318 | 3677 | 2018-11-11 | 2018-12-02 | TA, DIC |

## 3.2 Analytical methods – carbonate parameters

In 2014, the first year of the program, all three carbonate parameters (pH, TA and DIC) were analyzed for the Maritimes region at the Bedford Institute of Oceanography (BIO, Dartmouth, NS). In subsequent years, TA and pH were analyzed for the GSL region at the Maurice Lamontagne Institute (MLI, Mont-Joli, QC); DIC was added to the parameters analyzed in fall 2019, and pH was only measured in the spring of 2015 (Trip_Name "IML201516"). After 2014, TA and DIC were analyzed for the Maritimes (BIO) and NL (Northwest Atlantic Fisheries Centre – NAFC, St. John's,

NL) regions by their respective facilities. An overview of the carbonate parameters measured during each mission is provided in Table 1.

DIC (in $\mu\text{mol}\,\text{kg}^{-1}$ of seawater) is extracted as $CO_2$ by purging an acidified (1 M, 8.5 % phosphoric acid in excess) aliquot of water (warmed to 25 °C) with ultrahigh-purity (UHP) nitrogen gas using an automated sampling and gas extraction system. The dried gas, including the $CO_2$, is transferred and absorbed into a coulometric cell and analyzed by titration and photometric detection (Johnson et al., 1993). Various makes and models of auto-samplers and analyzers were used by the facilities in the different regions. Dupli-

| Year | Season | Region | Trip name | Stations | Bottles | Max depth | Start date | End date | Measured parameters |
|------|--------|--------|-----------|---------|---------|-----------|-----------|---------|---------------------|
| 2019 | Spring | GSL | IML2019040 | 1 | 188 | 332 | 2019-04-12 | 2019-12-13 | TA, pH |
| | | | IML201909 | 71 | 276 | 460 | 2019-05-26 | 2019-06-15 | TA, pH |
| | | MAR | COR2019001 | 42 | 295 | 3770 | 2019-04-09 | 2019-04-25 | TA, DIC |
| | | NL | TEL197 | 15 | 105 | 1197 | 2019-04-12 | 2019-04-18 | TA, DIC |
| | | | TEL199 | 1 | 7 | 100 | 2019-04-20 | 2019-04-20 | DIC |
| | | | TEL200 | 55 | 432 | 1207 | 2019-06-27 | 2019-07-13 | TA, DIC |
| | Summer | GSL | PER2019140 | 30 | 60 | 46 | 2019-07-23 | 2019-08-05 | TA, pH |
| | | | IML2019036 | 20 | 101 | 516 | 2019-08-15 | 2019-09-03 | TA, pH |
| | | | PER2019008 | 20 | 60 | 134 | 2019-08-15 | 2019-08-18 | TA, pH |
| | Fall | GSL | TEL2019201 | 35 | 126 | 372 | 2019-09-09 | 2019-10-01 | TA, DIC, pH |
| | | | PER2020019 | 5 | 14 | 89 | 2019-09-22 | 2019-10-07 | TA, DIC, pH |
| | | | IML2019049 | 45 | 228 | 470 | 2019-10-22 | 2019-11-06 | TA, DIC, pH |
| | | NL | COO001 | 43 | 306 | 1220 | 2019-11-17 | 2019-12-10 | TA, DIC |
| 2020 | Spring | GSL | IML2020040 | 1 | 120 | 332 | 2020-01-06 | 2020-12-09 | TA, DIC, pH |
| | Summer | NL | TEL210 | 43 | 313 | 1209 | 2020-07-14 | 2020-07-31 | TA, DIC |
| | | | AMU014 | 1 | 5 | 150 | 2020-08-11 | 2020-08-11 | TA, DIC |
| | Fall | GSL | IML2020028 | 39 | 206 | 470 | 2020-10-19 | 2020-10-30 | TA, DIC, pH |
| | | MAR | HUD2020063 | 34 | 255 | 2000 | 2020-10-04 | 2020-10-14 | TA, DIC |
| | | NL | HUD120 | 36 | 261 | 3668 | 2020-11-10 | 2020-12-01 | TA, DIC |

cate or triplicate sample measurements provided an analytical precision of 0.03 % at BIO and $< 0.1$ % at MLI and NAFC.

TA (in $\mu$mol kg$^{-1}$ of seawater) is measured by open-cell potentiometric titration (Dickson, 2010; Mintrop et al., 2000) with an automated sampling system. Each 50 mL (NAFC and BIO) or 104 mL (IML) sample, warmed to 25 °C, is titrated with 0.1 M hydrochloric acid to the Gran equivalence point (nonlinear curve-fitting method) using a computer-controlled Dosimat (Metrohm AG, USA) dispenser and combination glass electrode. In the NL region, aliquots of 0.1 M hydrochloric acid are added in a 25 °C thermostated jacket open cell while gently mixing the sample, and pH measurements are allowed to stabilize between successive readings. The analytical precision of TA at BIO was calculated using repeat analyses of bulk seawater and reported as $\pm 0.05$ %, while duplicate sample analyses revealed a precision of $< 0.1$ % at IML and NAFC.

Seawater pH, expressed using the total hydrogen ion scale (pH$_T$), was determined by spectrophotometry and calculated following the dye perturbation method (Clayton and Byrne, 1993; Dickson, 2010). Purified *m*-cresol purple solution (University of South Florida) was added to seawater held at $25 \pm 0.05$ °C in a 10 cm path-length quartz cell and mixed thoroughly. The ratio of blank-corrected absorbances measured at 434, 578 and 730 nm with a spectrophotometer (Agilent Technologies, USA) was used to determine pH$_T$. pH measurements are not corrected for dye perturbation, but the *m*-cresol dye is visually inspected each day and measurements are made to ensure its stability over time. Precision and accuracy, evaluated daily by repeat measurements of a tris(hydroxymethyl)aminomethane (TRIS) buffer solution (Andrew Dickson, Scripps Oceanographic Institution[CE4], San Diego, CA) prepared with a salinity of 30 (Millero, 1986), were typically $\pm 0.002$ pH units at BIO and $\pm 0.003$ pH units or better at IML.

All analytical methods were calibrated with a series of seawater certified reference materials (CRMs; Andrew Dickson[TS1], Scripps Institute of Oceanography, San Diego, CA), which allowed for performance evaluations of the various instruments and normalization of the TIC, TA and pH measurements. As the analyses were conducted in three different regions, we will consider the largest analytical error provided for all measurements: TA and DIC values will be $\pm 0.1$ % or $\pm 2.1$ $\mu$mol kg$^{-1}$ and pH will be $\pm 0.003$ units. These uncertainties are comparable to those suggested by Dickson (2010)

| Year | Season | Region | Trip name | Stations | Bottles | Max depth | Start date | End date | Measured parameters |
|------|--------|--------|-----------|----------|---------|-----------|------------|----------|---------------------|
| 2021 | Spring | GSL | IML2021011 | 1 | 150 | 332 | 2021-04-08 | 2021-12-16 | TA, pH |
| | | | IML2021012 | 41 | 218 | 466 | 2021-06-02 | 2021-06-12 | TA, pH |
| | | | IML2021014 | 30 | 59 | 397 | 2021-06-12 | 2021-06-22 | TA, pH |
| | | NL | TEL220 | 51 | 355 | 1204 | 2021-06-29 | 2021-07-19 | TA, DIC |
| | Summer | GSL | PER2021151 | 35 | 70 | 58 | 2021-07-06 | 2021-08-02 | TA, pH |
| | | | IML2021030 | 31 | 153 | 529 | 2021-07-27 | 2021-08-24 | TA, pH |
| | | | PER2021156 | 17 | 49 | 113 | 2021-08-30 | 2021-09-08 | TA, pH |
| | | | CJC2021222 | 36 | 104 | 352 | 2021-08-31 | 2021-09-26 | TA, pH |
| | | | PER2021021 | 2 | 6 | 92 | 2021-09-25 | 2021-09-25 | TA, pH |
| | Fall | GSL | IML2021044 | 50 | 307 | 469 | 2021-10-12 | 2021-10-26 | TA, pH |
| | | MAR | HUD2021185 | 63 | 409 | 4712 | 2021-09-17 | 2021-10-01 | TA, DIC |
| | | NL | TEL228 | 1 | 6 | 150 | 2021-12-20 | 2021-12-20 | TA, DIC |
| 2022 | Spring | GSL | IML202201 | 1 | 120 | 334 | 2022-04-27 | 2022-12-07 | TA, DIC, pH |
| | | | IML2022021 | 74 | 325 | 468 | 2022-06-06 | 2022-06-26 | TA, DIC, pH |
| | | | PER2022152 | 36 | 72 | 43 | 2022-06-30 | 2022-08-02 | TA, DIC, pH |
| | | MAR | AT4802 | 59 | 332 | 3775 | 2022-03-22 | 2022-04-04 | TA, DIC |
| | | NL | TEL229 | 1 | 6 | 150 | 2022-01-10 | 2022-01-10 | TA, DIC |
| | | | TEL227 | 1 | 6 | 150 | 2022-01-20 | 2022-01-20 | TA, DIC |
| | | | ATL001 | 57 | 427 | 4292 | 2022-04-10 | 2022-05-01 | TA, DIC |
| | Summer | GSL | IML2022039 | 25 | 157 | 521 | 2022-08-12 | 2022-09-14 | TA, DIC, pH |
| | | | PER202223330 | 2 | 6 | 96 | 2022-09-08 | 2022-09-08 | TA, DIC, pH |
| | | | CAR2022025 | 32 | 118 | 276 | 2022-09-13 | 2022-09-30 | TA, DIC, pH |
| | | | PER2022022 | 7 | 19 | 91 | 2022-09-28 | 2022-09-30 | TA, DIC, pH |
| | | NL | NED558 | 2 | 12 | 150 | 2022-07-01 | 2022-07-02 | TA, DIC |
| | | | PER004 | 1 | 6 | 150 | 2022-08-18 | 2022-08-18 | TA, DIC |
| | Fall | GSL | IML2022054 | 51 | 321 | 467 | 2022-10-25 | 2022-11-08 | TA, DIC, pH |
| | | NL | COO002 | 38 | 290 | 3676 | 2022-10-22 | 2022-11-08 | TA, DIC |

for modern analytical techniques using reference materials. All data flagged as suspect because of analytical error by each laboratory have been removed from the dataset.

## 3.3 Calculation of carbonate parameters

The aragonite saturation state ($\Omega_{arg}$), calcite saturation state ($\Omega_{cal}$, which is included in the dataset but not discussed in this report), $pH_{T,is}$ (total scale, in situ) and the partial pressure of $CO_2$ ($pCO_2$, in µatm) were calculated using the CO2SYS program (Lewis and Wallace, 1998) modified for Python (https://github.com/mvdh7/PyCO2SYS/tree/v1.2.

1, last access: 8 January 2023; Humphreys et al., 2020) as recommended for "best practices" by Orr et al. (2015). The dissociation constants ($K_1$ and $K_2$) of Mehrbach et al. (1973), as refit by Dickson and Millero (1987) (Mdm); the total boron constant from (Uppstrom, 1974); and the $KHSO_4$ constant from Dickson (1990) were also used as recommended for best practices (Chen et al., 2015; Dickson, 2010; Orr et al., 2015). Although it has been suggested that dissociation constants formulated for estuarine waters (Cai and Wang, 1998; Millero, 2010) should be used to avoid differences in the calculation of carbonate parameters at low salinity (Dinauer and Mucci, 2017), there is some evidence that

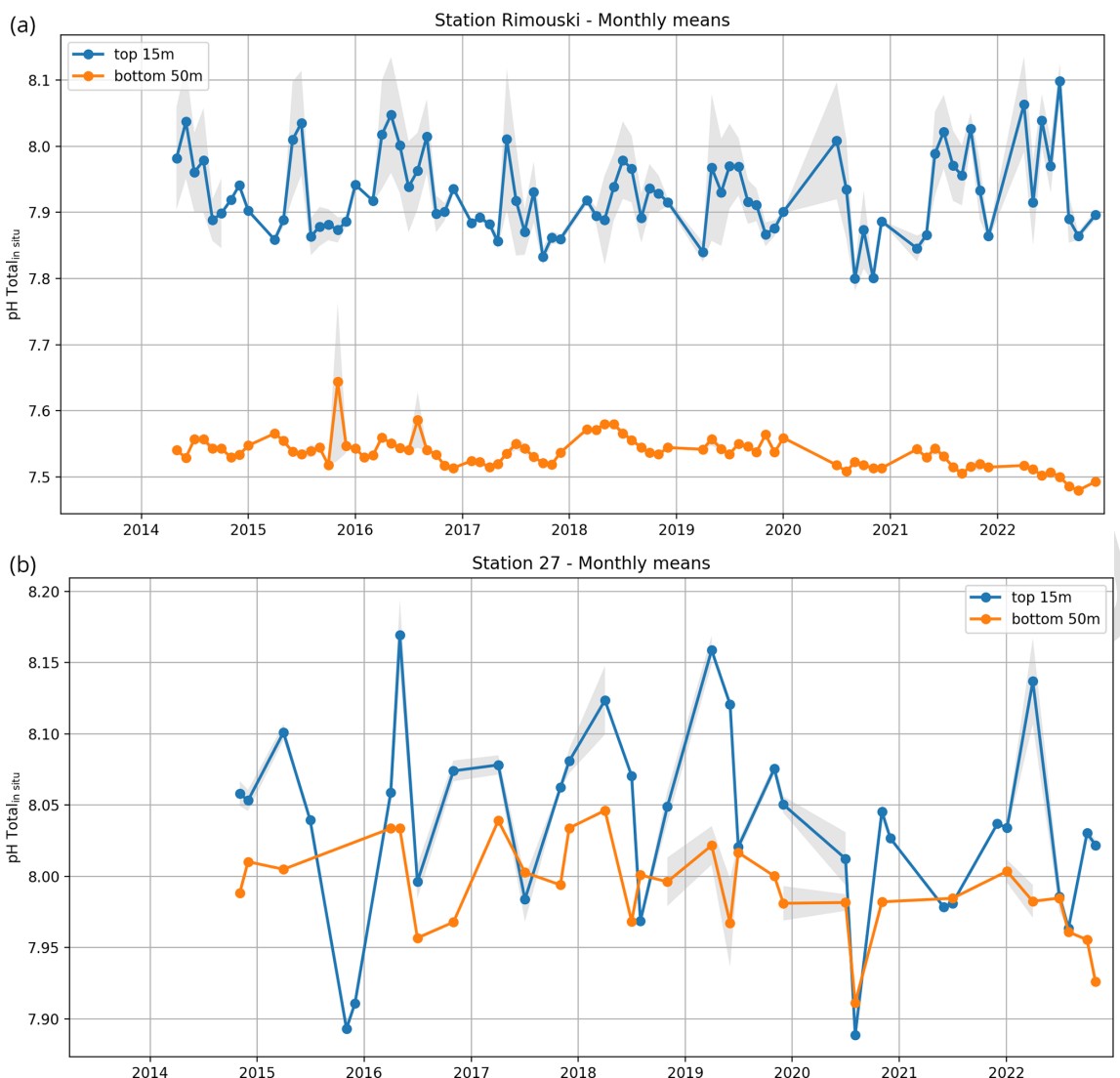

**Figure 4.** Monthly averages of pH data at Rimouski station **(a)** and Station 27 **(b)**. Surface observations (blue curves) are considered the average of the top 15 m, and bottom observations (orange curves) are the average of the bottom 50 m. The shaded area correspond to $\pm 0.5$ SD of the monthly distribution (when more than one visit per month is made). The tick marks on the horizontal axis correspond to the beginning of the year.

formulations other than those for best practices may produce discrepancies (Orr et al., 2015; Dinauer and Mucci, 2018). The dissociation constants by Mdm were formulated for salinities $> 20$. Samples with salinities $< 20$ constitute only 0.15 % of this dataset (1 % of the data from the GSL), and the differences in calculated carbonate parameters using the constants by Cai and Wang (1998) and Mdm are approximately double for samples with salinities $\geq 20$ compared with those with salinities $< 20$ (Table 2a). Therefore, the dissociation constants by Mdm were used in order to avoid discrepancies in the other 99.85 % of the dataset. The chosen combination of constants listed above was also suggested by Raimondi et al. (2019) because they generated the best internal consistency in their Labrador Sea (AR7W) dataset. However, their choice of constants provided less uncertainty than their choice of input parameters. The carbonate parameters in this dataset were calculated using the TA ($\mu$mol kg$^{-1}$ SW, where SW denotes seawater) provided by all regions and either DIC ($\mu$mol kg$^{-1}$ SW) or pH$_T$, which is region dependent. According to the results by Raimondi et al. (2019), the TA–DIC-based calculations of pH (e.g., NL and MAR regions) produce low accuracy but low uncertainty, and the TA–pH-based calculations of DIC (e.g., GSL dataset) provide the best consistency. The TA–pH pair will thus be used for calculating $p$CO$_2$ when both pH and DIC were collected in addition to TA (see Table 1). As the GSL region consistently measures the three carbonate equilibria parameters since the fall of 2019 (TA, pH and DIC; see Table 1), we esti-

mated the variability in the other derived parameters (in situ pH, $\Omega$ and $p\mathrm{CO}_2$) using the different combinations of pairs for this region between 2019 and 2020 as well as for 2022 (in 2021, only TA and pH were measured). Results are presented in Table 2d and discussed later in the text.

The spring 2015 GSL dataset (Trip_Name "IML201516") consists solely of pH data (in vitro at 25 °C). In order to obtain in situ values and other carbonate parameters, TA was estimated using the TA–salinity linear relationship (Cai et al., 2010; Fassbender et al., 2017) for $S > 20$. Using spring data at all depths and remaining years in the GSL (excluding the high-resolution Rimouski station data) led to the following fit: $\mathrm{TA} = 43.09 \times S + 808.31$ ($r^2 = 0.98$, $p < 0.001$; see Fig. 5b). The estimated freshwater TA end-member ($S = 0$) is $808.31\,\mathrm{\mu mol\,kg^{-1}}$. This value is lower than the measured values of $1204 \pm 99\,\mathrm{\mu mol\,kg^{-1}}$ or $1081 \pm 30.2\,\mathrm{\mu mol\,kg^{-1}}$ by Dinauer and Mucci (2017, 2018) near Québec City or the $1000\,\mathrm{\mu mol\,kg^{-1}}$ measured by Mucci et al. (2017) west of Saguenay Fjord. It is, however, higher than the $186\,\mathrm{\mu mol\,kg^{-1}}$ calculated by Dinauer and Mucci (2018) as a measure of the Saguenay Fjord and North Shore rivers and the $80\,\mathrm{\mu mol\,kg^{-1}}$ measured by Mucci et al. (2017) in the Saguenay River. Using a localized dataset for this calculation accounted for the low-salinity water exiting the St. Lawrence River. Considering the measurement error of up to 0.1 %, there was no significant difference in the predicted TA between the spring dataset and all seasons in the GSL. To estimate the degree to which the predicted TA had an effect on the calculation of the other carbonate parameters, the parameters for the spring 2015 GSL dataset were calculated in CO2SYS (TA–pH) with the maximum and minimum values of $\mathrm{TA} \pm 2\,\mathrm{SD}$ ($1.28\,\mathrm{\mu mol\,kg^{-1}}$), the TA–salinity regression uncertainty. This exercise shows the low sensitivity of this method, with an error of less than 0.03 % (Table 2c).

The calculated carbonate parameters were calibrated to in situ conditions using temperature (°C), salinity (psu) and pressure (dbar). The nutrient alkalinity parameters, represented by total soluble reactive phosphorus ($\mathrm{PO}_4$, in $\mathrm{\mu mol\,kg^{-1}}$ SW) and total soluble reactive silicate ($\mathrm{SiO}_3$, in $\mathrm{\mu mol\,kg^{-1}}$ SW), were used as additional information when available, as they contribute to the calculation of the total carbonate alkalinity (Orr et al., 2015). There can be a slight offset in calculated carbonate parameters when the nutrient data are absent, especially in high-concentration regions. This offset has been calculated for the Atlantic Zone by obtaining the difference in carbonate parameters calculated with and without nutrients (nutrients set to zero in CO2SYS; Fassbender et al., 2017). The calculation was performed on two datasets: the TA–DIC input pair from the NL and MAR regions and the TA–pH input pair from the GSL region. The results (Table 2b) indicate that all calculated carbonate parameters except for $p\mathrm{CO}_2$ will be slightly higher when nutrient data are lacking, with the TA–DIC input pair having the largest offset. The differences in the calculated pH (0.002 pH units) and DIC ($1.02\,\mathrm{\mu mol\,kg^{-1}}$) values are below the analytical

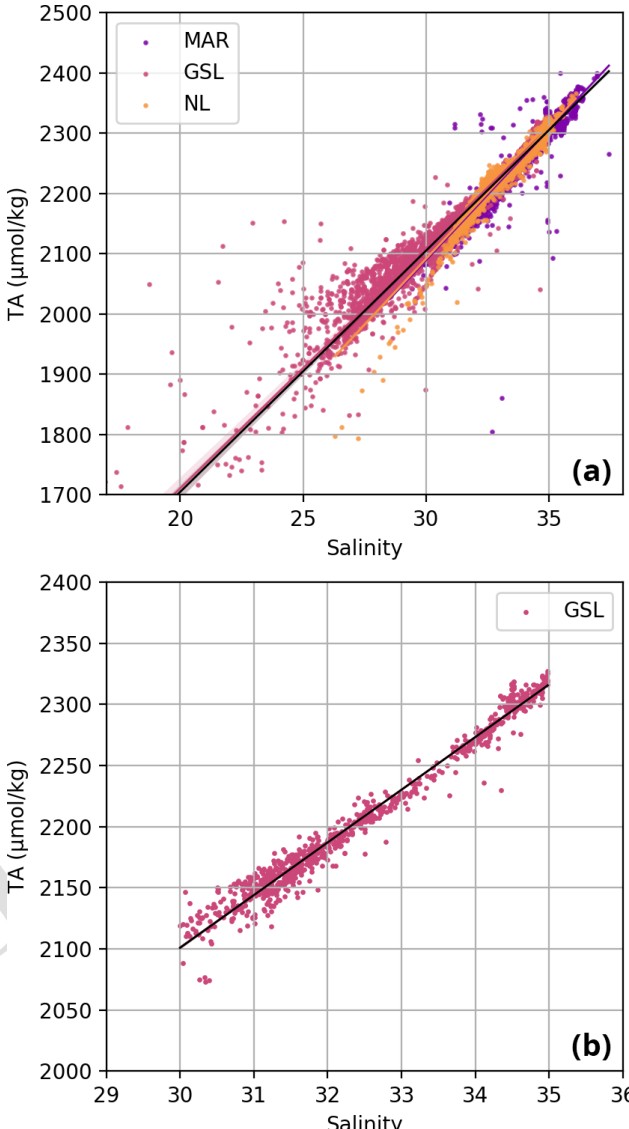

**Figure 5.** Scatterplot of total alkalinity (TA, in $\mathrm{\mu mol\,kg^{-1}}$) vs. salinity. In panel **(a)**, all samples (2014–2022) are a linear fit between TA and salinity and have been established independently for each region (coloured lines according to the legend) and for the entire dataset (black line). The latter TA–salinity relationship is $\mathrm{TA} = 40.05 \times S + 872.05$ (TS3 $r^2 = 0.94$, $p < 0.001$). In panel **(b)**, a linear relationship is also established for the GSL spring data with $S \geq 30$ only. This relationship between TA and salinity is $\mathrm{TA} = 43.09 \times S + 808.31$ (TS4 $r^2 = 0.98$, $p < 0.001$) and is used in Sect. 3.3 to correct for missing TA data.

uncertainty ($\pm 0.005$ pH units and $\pm 2.1\,\mathrm{\mu mol\,kg^{-1}}$, respectively). The mean saturation states and $p\mathrm{CO}_2$ have changed by < 0.6 % for the TA–DIC pair and < 0.1 % for the TA–pH pair. The larger nutrient–no nutrient difference with the TA–DIC pair may be due to the higher nutrient concentrations in the GSL region (> $17.3\,\mathrm{\mu M\,PO_4}$ and > $49.1\,\mathrm{\mu M\,SiO_3}$) or may reflect the input-parameter-propagated error as sug-

**Table 2.** Variability in calculated carbonate parameters using CO2SYS due to different sets of variables. Panel **(a)** shows the variability due to the choice of dissociation constants for low salinities (Mehrbach et al., 1973, as refit by Dickson and Millero, 1987, rather than Cai and Wang, 1998). Panel **(b)** displays variability due to the presence/absence of nutrients for the two sets of input pairs. Panel **(c)** shows the sensitivity of the calculated TA from a linear relationship with salinity (determined by multiplying the linear fit by twice its standard deviation). The absence of pH data at salinities < 20 is due to the fact that these low salinities are located in the GSL, which is the region that measures pH. Panel **(d)** presents the variability in the calculated carbonate parameters using CO2SYS with different carbonate equilibria pairs when all three parameters are available. We used data for the GSL region only, where most of these data occur. n/a represents not applicable CE5. TS2

| Difference (%) | pH | $\Omega_{arg}$ | $\Omega_{cal}$ | $p\mathrm{CO_2}$ | DIC | TA |
|---|---|---|---|---|---|---|
| **(a)** Choice of dissociation constant | | | | | | |
| $S_A < 20$ | 0.2 | 0.8 | 0.8 | 4 | 0.2 | 0 |
| $S_A \geq 20$ | −0.006 | 0.8 | 0.8 | −0.5 | −0.02 | < 0.001 |
| **(b)** Using nutrients in CO2SYS | | | | | | |
| TA–DIC (nutrients minus no nutrients) | −0.03 | −0.5 | −0.5 | −0.5 | 0 | 0 |
| TA–pH CE6 (nutrients minus CE7 no nutrients) | 0.003 | 0.05 | 0.05 | −0.05 | 0 | 0.08 |
| **(c)** Error when inferring TA from $S_A$ (spring 2015, GSL; $N = 493$) | | | | | | |
| TA–pH ($\pm 2\sigma$ on predicted TA) | 0 | ±0.03 | ±0.03 | ±0.03 | ±0.03 | ±0.03 |
| **(d)** Variability in carbonate equilibria pairs (GSL only, $S > 20$; $N = 2150$) | | | | | | |
| TA–DIC vs. TA–pH | 0.4 | 7 | 7 | −9 | −0.5 | n/a |
| TA–DIC vs. DIC–pH | 0.4 | 7 | 7 | −9 | n/a | 0.5 |
| DIC–pH vs. TA–pH | < 0.001 | −0.5 | −0.5 | −0.6 | −0.5 | −0.5 |

gested by Raimondi et al. (2019) and Dickson (2010), the latter of whom estimated the uncertainty in calculating the saturation state to be 3.7 % using the TA–pH and 1.7 % using TA–DIC. CE8

However, the entire dataset used here is provided for follow-up studies (see the "Data availability" section). This dataset includes the following physical, biogeochemical and carbonate parameters: temperature ($T$); salinity ($S$); dissolved oxygen (DO) concentration; oxygen saturation ($\mathrm{O_{2_{sat}}}$); nutrient data ($\mathrm{PO_4}$, $\mathrm{SiO_3}$ and $\mathrm{NO_3 + NO_2}$); measured and calculated (same as measured when available) TA and DIC; measured pH in the laboratory (when available); and calculated in situ pH, $p\mathrm{CO_2}$, $\Omega_{arg}$ and $\Omega_{cal}$.

## 4   Results and discussion

In the Atlantic Zone, the spatial and temporal variability in carbonate parameters reflects changes in both physical (temperature and salinity) and biological (plankton photosynthesis and respiration) parameters. While a complete description of the seasonal variability in the Atlantic Zone is difficult due to the timing of the surveys and large geographic areas of all three regions, some general description can be drawn.

In the following subsections, a summary of the physical and carbonate parameters is presented for different regions, seasons and depth ranges. Here, surface is defined as the shallowest sample but no deeper than 52 m, which is chosen to encompass sampling variations around the targeted 50 m depth, and the bottom is defined as the deepest sample be-

low 50 m. Due to some instances where near-bottom samples were not achieved (> 1200 m areas due to ship limitation, e.g., NL Slope), the following descriptions are restricted to depths shallower than 600 m. However, all of the data are available in the dataset.

A subset of the data (excluding nutrients and $\mathrm{O_2}$) is presented in Appendix A on maps of the surface and bottom waters of the Atlantic Zone for each sampling season of 2017, the most extensively sampled year to date (Figs. A1–A9). Each map represents one variable per depth per season. Seasons are defined as spring (the beginning of March to the end of June), summer (the beginning of July to 14 September) and fall (15 September to the end of December). The fall 2017 maps are presented in Figs. 6–8.

Depth profiles along a series of hydrographic sections from each region during the fall of 2017 are presented in Appendix B for the same set of variables. TS5 These sections include Seal Island (SI; Fig. B1), the Flemish Cap (FC; Fig. B2), Halifax (HL; Fig. B3), the Cabot Strait (CSL; Fig. B4) and the Laurentian Channel (LC; Fig. B5), comprising the line of stations from the centre of the Lower St. Lawrence Estuary through the Laurentian Channel to Cabot Strait (Fig. 3a). The results will be discussed as they are introduced.

### 4.1   Oceanographic context ($T$, $S$ and $\mathrm{O_{2_{sat}}}$)

A series of temperature and salinity maps (Figs. 6a–d, A1, A2) illustrate the physical oceanographic properties of the

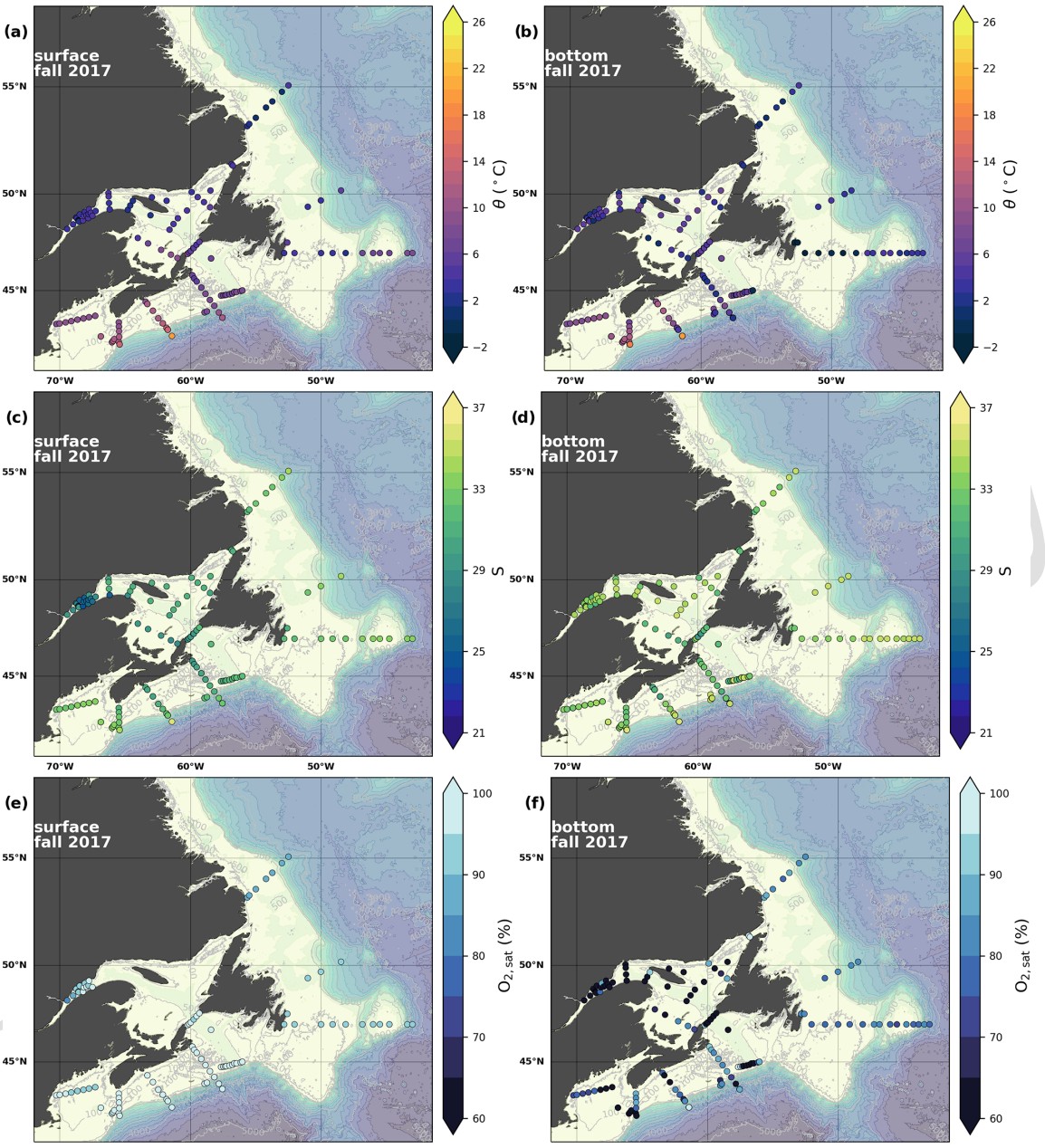

**Figure 6.** Maps of surface (uppermost sample $< 52$ m; **a**, **c**, **e**) and bottom (deepest sample $> 50$ m; **b**, **d**, **f**) temperature (**a**, **b**), salinity (**c**, **d**) and $O_{2_{\text{sat}}}$ (**e**, **f**) for fall 2017. The GSL surface $O_{2_{\text{sat}}}$ is represented by samples at 50 m.

Atlantic Zone. The Arctic waters of the coastal Labrador Current on the northern NL Shelf are characterized by low salinities and temperatures near freezing, whereas the Scotian Slope surface is characterized by warmer ($> 18\,°\text{C}$) and more saline ($> 33$) Gulf Stream waters (Fig. 2a). The lowest salinities ($< 27$) are found in the surface waters of the LSLE (ignoring the Saguenay Fjord and the USLE; see Fig. 3).

The moderately cold and saline surface waters along the southeastern NL Shelf and within the GSL consist of Arctic waters that have been modified by the Gulf Stream,

St. Lawrence River or warmed in summer. The freshwater exiting the LSLE can be traced at the surface along the coast of the Gaspé Peninsula into the southern GSL and Northumberland Strait (mean $S \sim 29$), where it warms significantly in summer (mean summer $T \sim 16\,°\text{C}$) (Figs. A1, A2). The GSL encompasses the greatest sea surface temperature differences between spring and summer; however, the largest differences are from spring to fall and are found on the Scotian Shelf ($> 15\,°\text{C}$). As the AZMP does not include carbonate winter sampling, the coldest temperatures in this dataset, down to

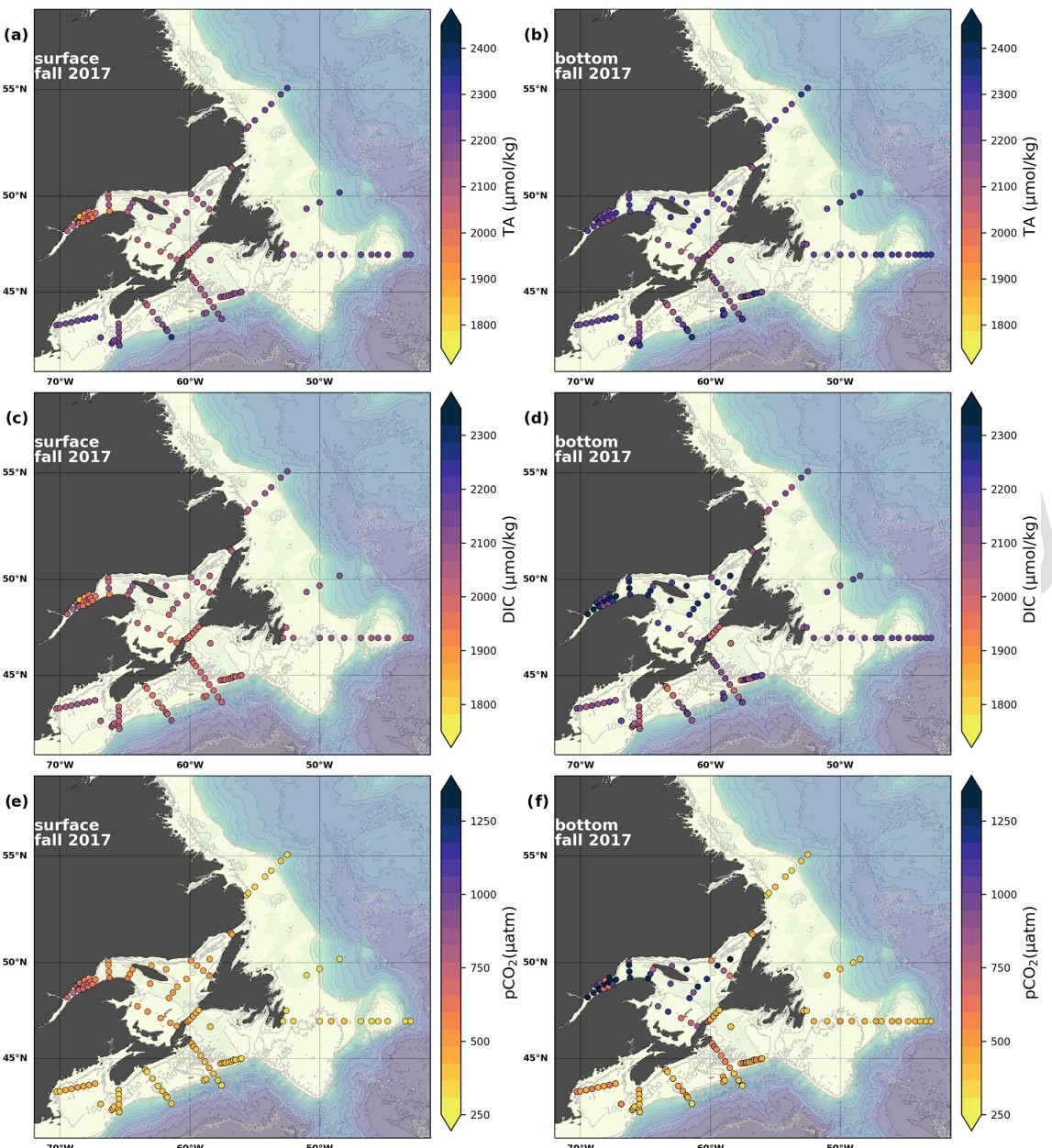

**Figure 7.** Maps of surface (uppermost sample $< 52$ m; **a**, **c**, **e**) and bottom (deepest sample $< 50$ m; **b**, **d**, **f**) TA (**a, b**), DIC (**c, d**) and $p$CO$_2$ (**e, f**) for fall 2017.

$-1.7\,^{\circ}$C, are located within the CIL, in the subsurface waters of the NL Shelf (e.g., hydrographic section SI; Fig. B1a) and in the GSL (e.g., hydrographic section LC; Fig. B5a). Warm, saline bottom waters influenced by the Gulf Stream can be traced along the southwest Scotian Shelf and Laurentian Channel into the GSL. Shallow bottom waters of the GSL can warm in summer and exceed $17\,^{\circ}$C (not shown).

The oxygen saturation (O$_{2_{\text{sat}}}$; see Sect. 3) is greatest at the surface, with mean regional values $> 90\,\%$ and supersaturation occurring during spring plankton blooms (Fig. 6e). Along the southern NL and Scotian outer shelf and slope, an oxygen minimum zone with O$_{2_{\text{sat}}}$ down to $45\,\%$ is centred around $250$ m (e.g., hydrographic section SESPB; Fig. 9). Mean bottom values on the NL Shelf range from nearly $100\,\%$ off Labrador to $\sim 55\,\%$ off southern NL. The O$_{2_{\text{sat}}}$ reaches supersaturation in the shallow bottom waters of the southern GSL and undersaturation in deeper areas associated with the Laurentian Channel. Hypoxia, which occurs at the bottom of the LSLE and deeper areas of the GSL, such as the Laurentian, Anticosti and Esquiman channels (Fig. 1; see also Gilbert et al., 2005), is shown as dark circles in Fig. 6f.

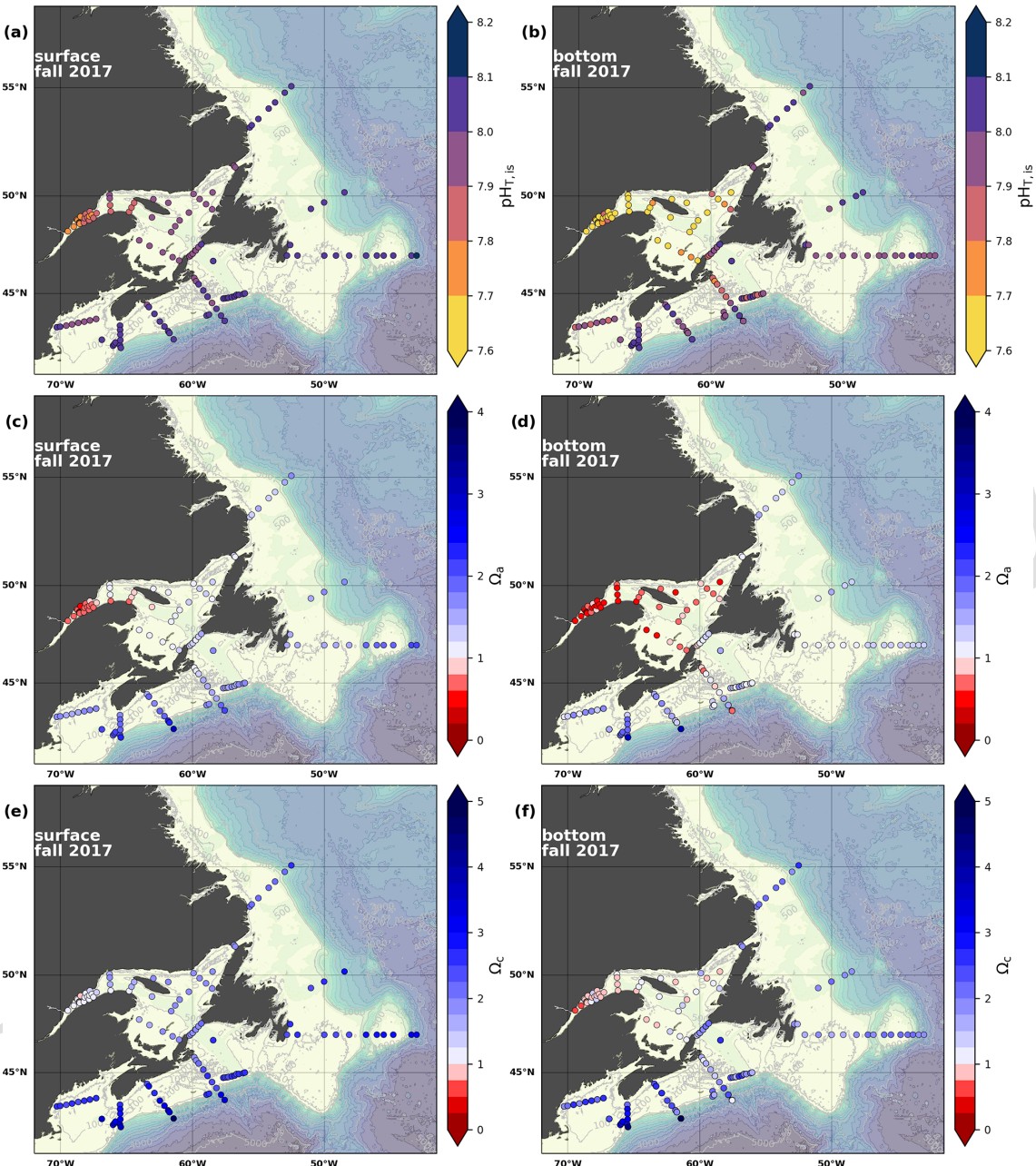

**Figure 8.** Maps of surface (uppermost sample < 52 m; **a, c, e**) and bottom (deepest sample > 50 m; **b, d, f**) pH **(a, b)**, $\Omega_{\mathrm{arg}}$ **(c, d)** and $\Omega_{\mathrm{cal}}$ **(e, f)** for fall 2017.

Changes in the physical environment partially drive the variability in the carbonate system. In recent years, the proportion of Labrador Current waters entering the deepest layers of the GSL in the Laurentian Channel has decreased at the expense of an increase in the North Atlantic Central Water (Jutras et al., 2020). This translated into a significant increase in the temperature of the bottom layers of the GSL and a concurrent decrease in dissolved oxygen and pH (DFO, 2023). On the NL Shelf, the variability in the cold and fresh Arctic-origin waters flowing southward along the coast influences

the carbonate system, with more acidic conditions expected during the colder and fresher years (Cyr et al., 2022b). On the Scotian Shelf, warmer and saltier conditions usually imply that the system is less prone to the undersaturation of the carbonate saturation states, but the changes in water masses' compositions may influence the interannual variability in carbonate parameters. In the next subsection, we describe the carbonate system and make links, when possible, with the oceanography of the region.

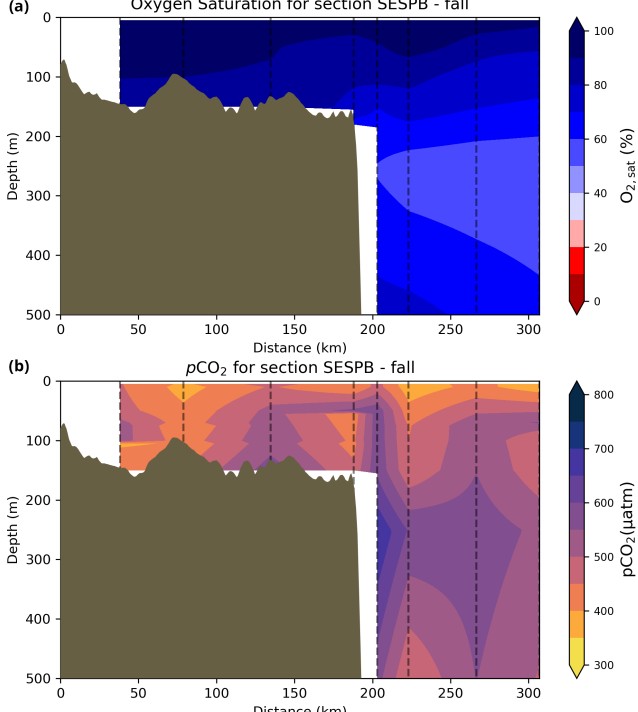

**Figure 9.** Depth–distance (the $x$ axis is the distance in kilometres from first station) contours along a standard section of the Southeast St. Pierre Bank (SESPB) off the southern Newfoundland Slope (2015–2018 average) for the **(a)** oxygen minimum and **(b)** $p\text{CO}_2$. The dashed lines represent positions of the casts where the samples were collected.

## 4.2 Carbonate parameters (TA, DIC, pH and $\Omega_{\text{arg}}$)

This section comprises a brief description of the spatial and temporal variations in the carbonate parameters of the Atlantic Zone. Except for 2015, the values and seasonal cycles among annual surveys are relatively comparable. For this reason, we chose the year 2017, one of the most complete year to date, both seasonally and spatially. Thus, in the following, descriptions made of the Atlantic Zone refer to maps and sections from 2017 (Figs. 6–8, A1–A9). Comparing seasonal changes (spring, summer and fall) in the Atlantic Zone or changes by region is difficult due to the surveying protocols established in each region. The hydrographic sections FC and BB on the NL Shelf (Fig. 3a) are the only ones planned for the three seasons every year; however, the station separation is greater along BB, and the resolution is consequently poorer. Although the standard AZMP GSL sections are only surveyed in spring and fall, the summer groundfish surveys between 2017 and 2022 provide enough data to evaluate some seasonal trends in the GSL (see below).

### 4.2.1 Total alkalinity

The linear relationship between TA and $S$ for the Atlantic Zone provides an $r^2$ value of 0.94 (TA $= 40.34 \times S + 893.64$, $p < 0.001$; Figs. 5), and TA values consequently follow the same spatial pattern as $S$ (Figs. 6c, d, 7a, b, A2, A4). The lowest TA concentrations are located in the surface waters of the LSLE and GSL due to freshwater outflow from the LSLE (the TA value at $S = 0$ is $< 1200\,\mu\text{mol}\,\text{kg}^{-1}$; Sect. 4.2). Lower TA values are also observed exiting the GSL along the southwestern Cabot Strait onto the Scotian Shelf (Fig. 7a).

The surface TA is also low along the NL coast which is associated with coastal runoff and the fresh coastal Labrador Current. The largest TA values, reaching nearly as high as $2400\,\mu\text{mol}\,\text{kg}^{-1}$, are found within the upper 150 m of the southern NL and within the Scotian shelf Slope Water. Following the seasonal cycle of salinity, TA generally decreases from spring to summer/fall on the NL/Scotian shelves, reflecting the respective continued freshening of the NL coast (Cyr and Galbraith, 2021) and the freshwater export from the LSLE exiting the Gulf of St. Lawrence at Cabot Strait (Dever et al., 2016). Although the TA of the Atlantic Zone bottom waters follows a similar spatial pattern to that at the surface, the values are higher and the range is much narrower (Fig. 7b). The mean shelf bottom ($< 600$ m) TA is $2241\,\mu\text{mol}\,\text{kg}^{-1}$, with the lowest values in the USLE and shallow southern GSL ($\sim 2000\,\mu\text{mol}\,\text{kg}^{-1}$). The highest TA values of the Atlantic Zone are in the warm Slope Water ($\sim 2300\,\mu\text{mol}\,\text{kg}^{-1}$), in deep waters of the Scotian Slope CE9 and GSL.

### 4.2.2 Dissolved inorganic carbon

The highest DIC concentrations are located in the deepest areas ($> 275$ m) of the LSLE and GSL, specifically the Laurentian, Anticosti and Esquiman channels, with maximum values of just over $2300\,\mu\text{mol}\,\text{kg}^{-1}$ (Figs. 7d, A5). Moderately high concentrations are observed deep offshore along the NL (up to $2264\,\mu\text{mol}\,\text{kg}^{-1}$) and Scotian (up to $2244\,\mu\text{mol}\,\text{kg}^{-1}$) shelves and slopes. These values decrease on the shelf toward the coasts, and the lowest values are in the shallow areas of the shelf under freshwater influence (down to $\sim 2000\,\mu\text{mol}\,\text{kg}^{-1}$ in the USLE). The accumulation of DIC within shelf bottom waters in fall is due to the remineralization of organic matter sinking to the bottom after spring and summer plankton blooms, with the highest values in the poorly ventilated deep GSL waters. The surface water has significantly lower DIC values, with the lowest concentrations occurring in the St. Lawrence Estuary as well as in the southern GSL and the coastal stations along the northern NL Shelf (Fig. 7c).

In southern NL and the eastern GSL, the DIC decreases from spring to summer as it is consumed during the plankton bloom, and it then increases from summer to fall dur-

ing the remineralization of that produced organic matter. The decrease in values from spring to fall along the Scotian Shelf reflects the transition from high winter DIC values due to remineralization, cold temperatures (increased solubility) and mixing with deep DIC enriched water during winter convection to lower values in fall due to photosynthesis and stratification.

### 4.2.3 Partial pressure of $CO_2$

The lowest $p\mathrm{CO_2}$ values (down to $\sim 200\,\mu\mathrm{atm}$) are located on the Newfoundland and Labrador Shelf, particularly in the coldest northern surface waters (Figs. 7e, A6). $p\mathrm{CO_2}$ generally increase from the surface to the bottom and from spring to fall as plankton from the spring bloom are consumed (surface/spring) and then remineralized (bottom/fall). Higher temperatures also influence the seasonal and southward increase in $p\mathrm{CO_2}$. Therefore, slightly higher values are located in the bottom waters of the southern NL and Scotian shelves and slopes, reaching a maximum ($\sim 600\,\mu\mathrm{atm}$) at the $O_2$ minimum (Fig. 9). Coincident with DIC, the highest $p\mathrm{CO_2}$ values are located in the deepest areas of the LSLE and GSL, with values $> 1500\,\mu\mathrm{atm}$ (Fig. B5). This $CO_2$, which accumulates at depth due to poor ventilation in the GSL, is produced by the remineralization of organic matter.

However, this study highlights some sensitivity in the choice of the carbonate equilibria parameters employed to derive $p\mathrm{CO_2}$ using CO2SYS (Table 2d). More precisely, results from the GSL (excluding the Saguenay), where TA, DIC and pH are measured ($N = 2150$), suggest that using the TA–DIC pair would lead to an almost 9 % decrease in the $p\mathrm{CO_2}$ estimate on average compared with using TA–pH or DIC–pH CE10 (with 95 % of the variation within the $[-27\,\%, +2\,\%]$ range). These results advocate for a direct measurement of $p\mathrm{CO_2}$ when possible and confirm the cautiousness needed when using derived values (Golub et al., 2017).

### 4.2.4 pH

In 2017, the pH of the Atlantic Zone ranged from approximately 7.5 in the LSLE and GSL bottom waters (and lower in the Saguenay River) to 8.3 in the surface waters off Labrador, with the highest values generally found along the shelf once the sea ice had melted in summer (Figs. 8a–b, A7). The plankton bloom, which consumes $CO_2$, is concurrent with the ice retreat, which increases the pH. Stations with a high pH are often located away from shore with a higher $T$, $S$ and TA (Figs. 6a–c, 8a) and are less influenced by coastal processes. A high pH is also observed during most annual spring surveys along the Scotian Shelf (particularly in 2017). These high pH values may be associated with the spring plankton bloom. The surface water pH is relatively uniform on the NL Shelf (range of [8.06, 8.10]), whereas it varies between 7.88 and 8.08 on the Scotian Shelf and between 7.74 and 7.98 for the GSL. The decreased GSL surface pH is influenced by the

freshwater exiting the LSLE (area of lowest surface pH) and high concentrations of plankton which consume $CO_2$ (pH increases) and are subsequently remineralized (pH decreases).

The bottom waters of the GSL, specifically the LSLE and northern and deeper areas of the GSL, including the Laurentian, Anticosti and Esquiman channels, contain the lowest pH (down to 7.5 in 2017) in the Atlantic Zone (Fig. 8b). These areas are also associated with the lowest $O_{2_{\mathrm{sat}}}$ ($\sim 16\,\%$ in 2017; Fig. 6f). Due to the estuarine nature of the GSL, its bottom waters have restricted ventilation, allowing for the accumulation of $CO_2$ and a reduction in the pH produced by the respiration of organic matter in this highly productive environment. The highest bottom pH values are located within the cold, shallow bottom waters of the GSL and in the coldest waters of the NL Shelf. The bottom pH of the Atlantic Zone generally decreases throughout the year, likely due to the remineralization of organic matter (Fig. A7). The spatial and temporal distribution of pH appears to be inversely correlated with $p\mathrm{CO_2}$, which describes a pattern of the photosynthesis and subsequent remineralization of plankton.

Data from the high-resolution Rimouski station and Station 27 (Fig. 4) show the spatial differences between the NL Shelf and the GSL as well as the interannual variability in pH in these regions. Station 27 exhibits a higher pH in general with little difference between the surface and the bottom, although interannual variations are large. At Rimouski station, a significant difference in pH exists between the surface and the bottom. There is also a documented significant decrease in pH in the bottom waters of the LSLE where Rimouski station is located, which reached a record low of 7.48 in October 2022 (Fig. 4a, orange line). This decrease in pH was accompanied by an increase in temperature at depth caused by the progression of warm waters in the deep channels of the GSL decrease and a decrease in dissolved oxygen saturation to unprecedented levels ($< 10\,\%$; not shown). Since 2018, these observations have been reported as part of the AZMP annual reports (e.g., DFO, 2023).

### 4.2.5 Saturation state relative to aragonite ($\Omega_{\mathrm{arg}}$)

The Atlantic Zone $\Omega_{\mathrm{arg}}$ ranges from 0.5 in the LSLE and GSL bottom waters (lower values in the Saguenay River) to 3.7 in the surface waters of the southwest Scotian Slope (Figs. 8c–d, A8). These highest $\Omega_{\mathrm{arg}}$ surface waters are most influenced by the warm and saline Gulf Stream waters (Fig. 6a–c, 8c; southernmost stations). $\Omega_{\mathrm{arg}}$ values $> 2.0$ are also located in the surface waters of the outer NL region CE11. On average, these Scotian and NL waters increase in $\Omega_{\mathrm{arg}}$ from spring to fall (Fig. A8), corresponding to the seasonal temperature increase and the decrease in $S$ and TA (Figs. A1, A2, A4). Excluding the LSLE, the surface GSL $\Omega_{\mathrm{arg}}$ values are similar to those from the southern NL and northeastern Scotian shelves ($\sim 1.5$–2.0). The surface waters of the USLE are undersaturated ($\Omega_{\mathrm{arg}} < 1$) with respect to aragonite (and, in some instances, calcite) due to low TA and DIC (Figs. A4,

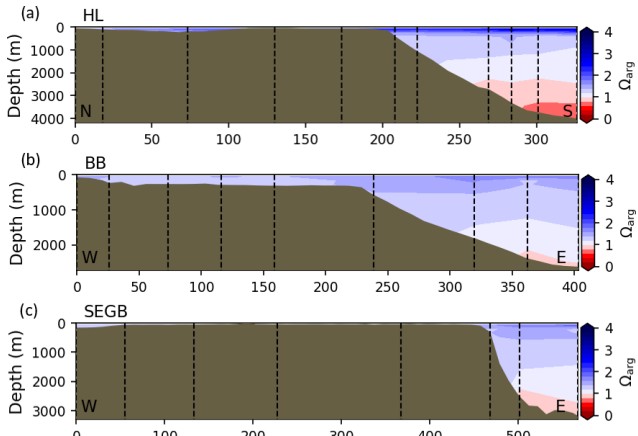

**Figure 10.** Depth profiles along standard sections in the Atlantic Zone for the fall of 2017. The aragonite saturation horizon ($\Omega_{arg} = 1$) is located between $\sim 2200$ and $2300\,\mathrm{m}$ along sections HL **(a)**, BB **(b)** and SEGB **(c)** above the Scotian and NL slopes.

A5, A8, A9) and can be traced into the GSL along the north coast of Gaspé. The bottom waters of the GSL, specifically the LSLE and northern and deeper areas of the GSL, including the Laurentian, Anticosti and Esquiman channels, are undersaturated with respect to aragonite (Fig. 8d).

Most of the bottom GSL samples ($> 90\,\%$) are undersaturated, with the few saturated samples in shallow areas or along the Cabot Strait section. The poorly ventilated GSL allows for the accumulation of DIC produced by the decomposition of organic matter in this highly productive environment. The high DIC, coupled with a low buffering capacity (TA), produced the undersaturated benthic conditions. Undersaturated waters also occur on the southeast Scotian Shelf, where GSL bottom water exits through Cabot Strait, as well as on the southern (colder) NL Shelf. These conditions are present during the summer (NL) and fall (MAR and NL) surveys due to the remineralization of the spring/summer plankton bloom. Although undersaturation is present, the mean $\Omega_{arg}$ on the NL and Scotian shelves is $\sim 1.3$. The deep Scotian and NL slopes are also undersaturated, and the saturation horizon ($\Omega_{arg} = 1$) is between 2200 and $2300\,\mathrm{m}$ (Fig. 10). In the Labrador Sea, Azetsu-Scott et al. (2010) demonstrated that this horizon corresponds to the depth where the Labrador Sea Water overlies the North East Atlantic Deep Water. Therefore, the depth of the saturation horizon along the Labrador Slope can fluctuate with time depending on the deep winter convection in the Labrador Sea (Azetsu-Scott et al., 2010; Yashayaev and Loder, 2017).

Similar to $p\mathrm{CO_2}$, some variability exists between $\Omega$ parameters derived using CO2SYS with different carbonate equilibria pairs (Table 2d). In this case, using the TA–DIC pair would lead to about a 7 % increase in the $\Omega$ estimate on average compared with using TA–pH or DIC–pH (with 95 % of the variation within the [−1 %, 20 %] range). While this variability is slightly less than for $p\mathrm{CO_2}$, care must be taken in the interpretation of the results provided here.

## 4.3 General overview of physical and biogeochemical interactions

An overview of the Atlantic Zone oceanographic setting and a description of the different carbonate parameters are provided in Sect. 4.1 and 4.2, respectively. A large range of values and seasonal variations were found for the different parameters presented. These elements are now discussed together in a broader regional and seasonal context.

The surface of the Atlantic Zone is characterized by cold (Arctic) and warm (Atlantic) waters that undergo a seasonal warming and freshening. This seasonal freshening generally lowers TA and, in turn, the capability of the system to buffer acidification. On the other hand, seasonal dynamics of phytoplankton blooms generally decrease inorganic carbon (e.g., DIC) in surface waters during the spring/summer as a result of photosynthesis. At depth, these values generally increase from spring to fall as plankton from the spring bloom are degraded and inorganic material is remineralized near the bottom.

The lowest $p\mathrm{CO_2}$ values are located in the cold coastal Labrador Current, whereas the highest values are found in the fresh waters of the LSLE. The lowest pH values are found in the GSL and the LSLE. This is especially true for the southern portion of the gulf and in the deep ($> 300\,\mathrm{m}$) layers of the Laurentian Channel. The most acidic waters of the Atlantic Zone are found in the Lower St. Lawrence Estuary, where the pH decreased by 0.2–0.3 between the 1930s and the early 2000s; this region is also affected by severe hypoxia (Mucci et al., 2011). These low and undersaturated conditions, which cover a large part of the Atlantic Zone, are concerning and unfavourable for calcifying organisms.

$\Omega_{arg}$ increases at the surface and in shallow areas of the shelf from spring to fall at most MAR and NL stations as a consequence of the seasonal temperature increase and the decrease in $S$ and TA coupled with the high DIC of respired organic matter (see also Shadwick et al., 2011a, b, for a description of the $\mathrm{CO_2}$ system seasonal cycle on the Scotian Shelf). Most of the bottom GSL is also undersaturated with respect to aragonite ($\Omega_{arg} < 1$), except for a few saturated samples in shallow areas and along the Cabot Strait section. Consistent with Azetsu-Scott et al. (2010), low $\Omega_{arg}$ values are found in the cold and fresh (thus lower TA) Arctic-origin waters of the coastal Labrador Current. These cold, acidic waters are a prominent feature of the Atlantic Zone, as they contribute to the LSW formation southwest of the Grand Banks. The LSW, which has a $\Omega_{arg} < 2$, partly forms the deep waters of the Laurentian Channel in the GSL (Gilbert et al., 2005) and then continues southward on the Scotian Slope and along the northeastern US coast. These low $\Omega_{arg}$ values can be traced to the deep Gulf of Maine. As the water temperature increases during the journey towards the south-

ern US, $\Omega_{arg}$ increases, reaching $\sim 4.5$ in the Gulf of Mexico (Wanninkhof et al., 2015).

This study demonstrates clear links between the physical environment and the carbonate system in the Atlantic Zone. Waters around Atlantic Canada have warmed over the last century as a consequence of anthropogenic climate change, and this trend is expected to continue in the future (Greenan et al., 2019). In this context, it is crucial to continue to monitor the physical and biogeochemical environment in the Atlantic Zone. While the 9-year time series presented here is relatively short, the data presented generally support other studies suggesting that the rate of change in ocean acidification and deoxygenation in Atlantic Canada may be greater than the global average (Claret et al., 2018; Bernier et al., 2018). The situation is especially dramatic in the GSL where the warm waters entering the deepest layers of the GSL have significantly decreased the dissolved oxygen and pH since about 2019 (Fig. 4; see also DFO, 2023).

## 5    Data availability

The full dataset of measured and derived parameters is available from the Federated Research Data Repository: https://doi.org/10.20383/102.0673 (Cyr et al., 2022a). It consists of a single comma-separated values (CSV) file named "AZMP_carbon.csv". This file contains 17 025 lines, each corresponding to a discrete sample. Of these lines, 15 683 contain the full suite of parameters derived using CO2SYS, whereas the remaining 1342 samples only have one of the three carbonate parameters (TA, TIC or pH) available. A brief description of the columns in this file is provided here (see Sect. 3 for a detailed description of the quantities):

- *Timestamp* – date and time (UTC) that the sample was collected in "yyyy-mm-dd HH:MM:SS" format;

- *Region* – geographical region were the sample was collected (e.g., "GSL", "MAR" or "NL");

- *Trip_Name* – name of the seagoing mission (convention varies between regions);

- *Station_Name* – hydrographic station names (e.g., "TESL3" or "STN27");

- *Latitude_(degNorth)* – latitude of the sampling location in decimal format (in $^\circ$ N);

- *Longitude_(degEast)* – longitude of the sampling location in decimal format (in $^\circ$ E);

- *Depth_(dbar)* – depth of the sampling (in dbar), although, in some rare circumstances, the nominal depth of the sample was used when the CTD depth was missing;

- *Temperature_(degC)* – temperature (in $^\circ$C) using the International Temperature Scale of 1990 (ITS-90);

- *Salinity_(psu)* – salinity expressed on the practical salinity scale (psu, unitless);

- *Dissolved_Oxygen_(mL/L)* – dissolved oxygen concentration (in $mL\,L^{-1}$) obtained using either Winkler titration or a DO sensor calibrated in situ with Winkler titration;

- *Nitrate_Concentration_(mmol/m3)* – nitrate (and nitrite) concentration (in $mmol\,m^{-3}$) determined in the laboratory with an auto-analyzer;

- *Phosphate_Concentration_(mmol/m3)* – phosphate concentration (in $mmol\,m^{-3}$) determined in the laboratory with an auto-analyzer;

- *Silicate_Concentration_(mmol/m3)* – silicate concentration (in $mmol\,m^{-3}$) determined in the laboratory with an auto-analyzer;

- *Total_Alkalinity_(umol/kg)* – total alkalinity (TA, in $\mu mol\,kg^{-1}$) measured or derived using CO2SYS (no missing values);

- *Inorganic_Carbon_(umol/kg)* – inorganic carbon (DIC, in $\mu mol\,kg^{-1}$) measured or derived using CO2SYS (no missing values);

- *pH_tot_(total_scale)* – in situ pH on the total scale (unitless) derived using CO2SYS (no missing values);

- *Omega_Aragonite_(unitless)* – saturation state relative to aragonite ($\Omega_{arg}$, unitless);

- *Omega_Calcite_(unitless)* – saturation state relative to calcite ($\Omega_{cal}$, unitless);

- *pCO2_(uatm)* – partial pressure of $CO_2$ derived using CO2SYS ($pCO_2$, in $\mu atm$);

- *Oxygen_Saturation_(%)* – dissolved oxygen saturation (in %);

- *pH_lab_(total_scale)* – pH measured in the laboratory (total scale, unitless), although the absence of a value signifies that this parameter was not measured;

- *pH_lab_temp_(degC)* – temperature at which the pH was measured in the laboratory (in $^\circ$C);

- *Total_Alkalinity_Measured_(umol/kg)* – total alkalinity (TA, in $\mu mol\,kg^{-1}$) measured in the laboratory, although the absence of value signifies that this parameter was not measured;

- *Inorganic_Carbon_measured_(umol/kg)* – inorganic carbon (DIC, in $\mu mol\,kg^{-1}$) measured in the laboratory, although the absence of value signifies that this parameter was not measured.

Note that the majority of the carbonate parameter data presented here have also been archived with extended meta-data on the Ocean Carbon Data System (OCADS): https://www.ncei.noaa.gov/access/ocean-carbon-data-system/ (last access: 13 July 2021). Users are invited to retrieve these data for a more in-depth analysis of the Atlantic Zone carbonate system.

## 6 Conclusion

As a result of increasing atmospheric $CO_2$ uptake, the ocean has undergone acidification during the 20th century and is projected to continue to acidify through the 21st century (Pörtner et al., 2019). Observation programs, such as the AZMP, are essential to monitor these changes and their consequences on ecosystems (e.g., Tilbrook et al., 2019). Because in situ observations are often scattered in time and space, ocean biogeochemical models are another important tool available to improve our understanding of seasonal, interannual and climatic variations in ocean acidification. Although the field of biogeochemical modelling is rapidly developing, the availability of biogeochemical data at meaningful spatial and temporal resolutions appears to be a considerable challenge that hinders the implementation or validation of such models at regional scales (Fennel et al., 2019; Capotondi et al., 2019; Pilcher et al., 2019; Lavoie et al., 2021). Thus, in addition to providing a baseline of carbonate parameters, a comprehensive overview of carbonate parameters, such as the one provided here, is a necessary and useful contribution to the modelling community.

Using the $0.5 \pm 0.2$ decrease in $\Omega_{arg}$ in the surface ocean by the year 2100 as suggested by Bates et al. (2009), it appears that ocean acidification may impact parts of the ecosystem within the next century. As the climate changes, however, the Atlantic Zone will be influenced by changes in temperature, salinity, ocean currents, nutrients and productivity, all of which will contribute to regional changes in pH and saturation state. While this study provides a useful baseline for the Atlantic Zone, incorporating these data into biogeochemical models is a necessary step in order to examine the regional response to ocean acidification and the future of the Atlantic Canadian aquaculture and fishing industries (Lavoie et al., 2020; Siedlecki et al., 2021).

## Appendix A: Seasonal maps of various parameters during 2017.

Here, we present a series of seasonal maps (spring, summer and fall) of various physicochemical parameters collected as part of the AZMP during 2017, one of the most complete years of sampling. These parameters are as follows: temperature, salinity, $O_{2_{sat}}$, TA, TIC, $pCO_2$, pH, $\Omega_{arg}$ and $\Omega_{cal}$. They are presented in Figs. A1 to A9, respectively. Each figure has six panels: the top row is the surface (uppermost sample $< 52$ m) and the bottom row presents the near-bottom (deepest sample $> 50$ m) conditions. The spring, summer and fall seasons are presented from left to right.

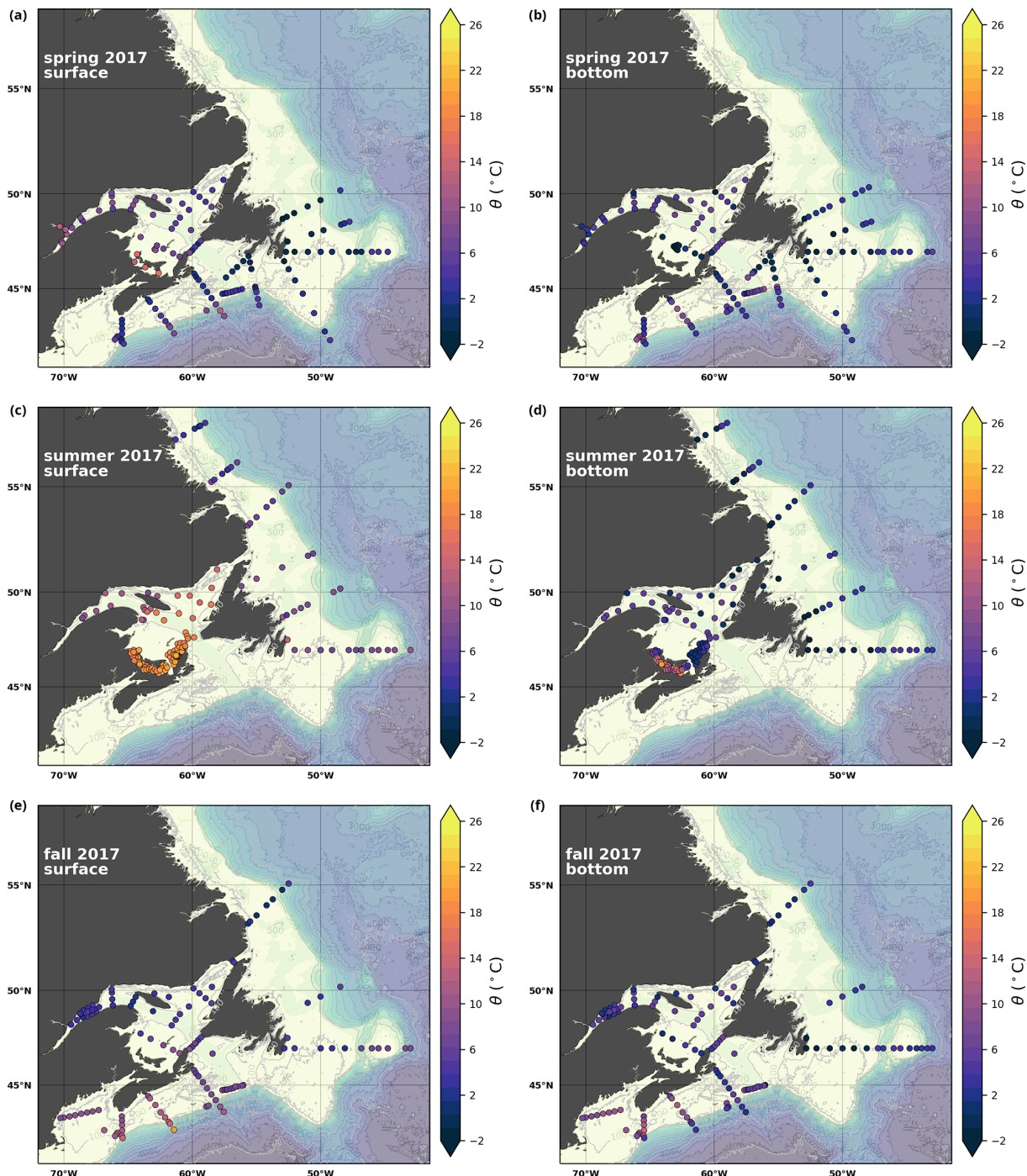

**Figure A1.** Surface (uppermost sample < 52 m; **a, c, e**) and near-bottom (deepest sample > 50 m; **b, d, f**) temperature sampled during the spring (**a, b**), summer (**c, d**) and fall (**e, f**) AZMP surveys. TS6

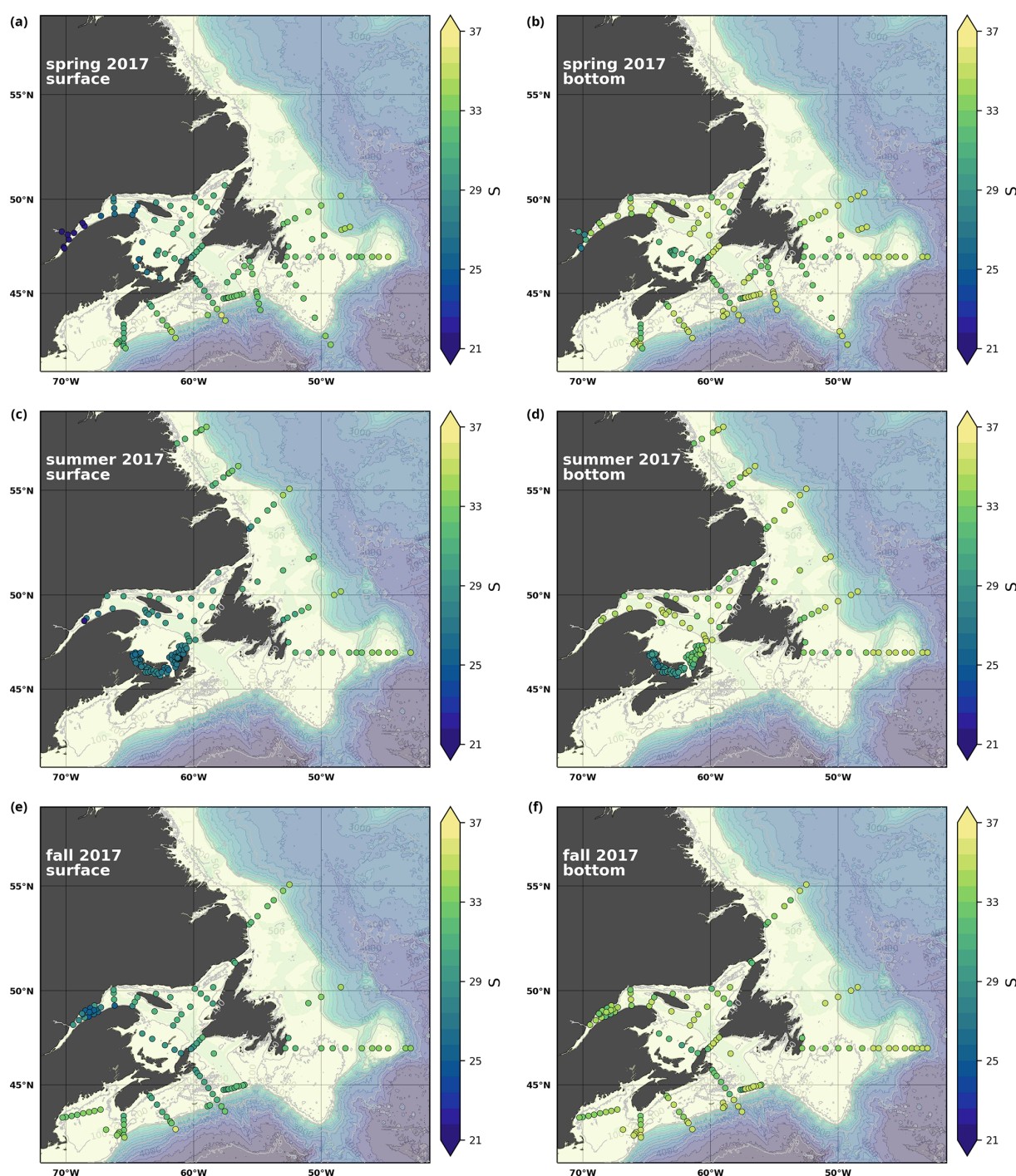

**Figure A2.** Same as in Fig. A1 but for salinity.

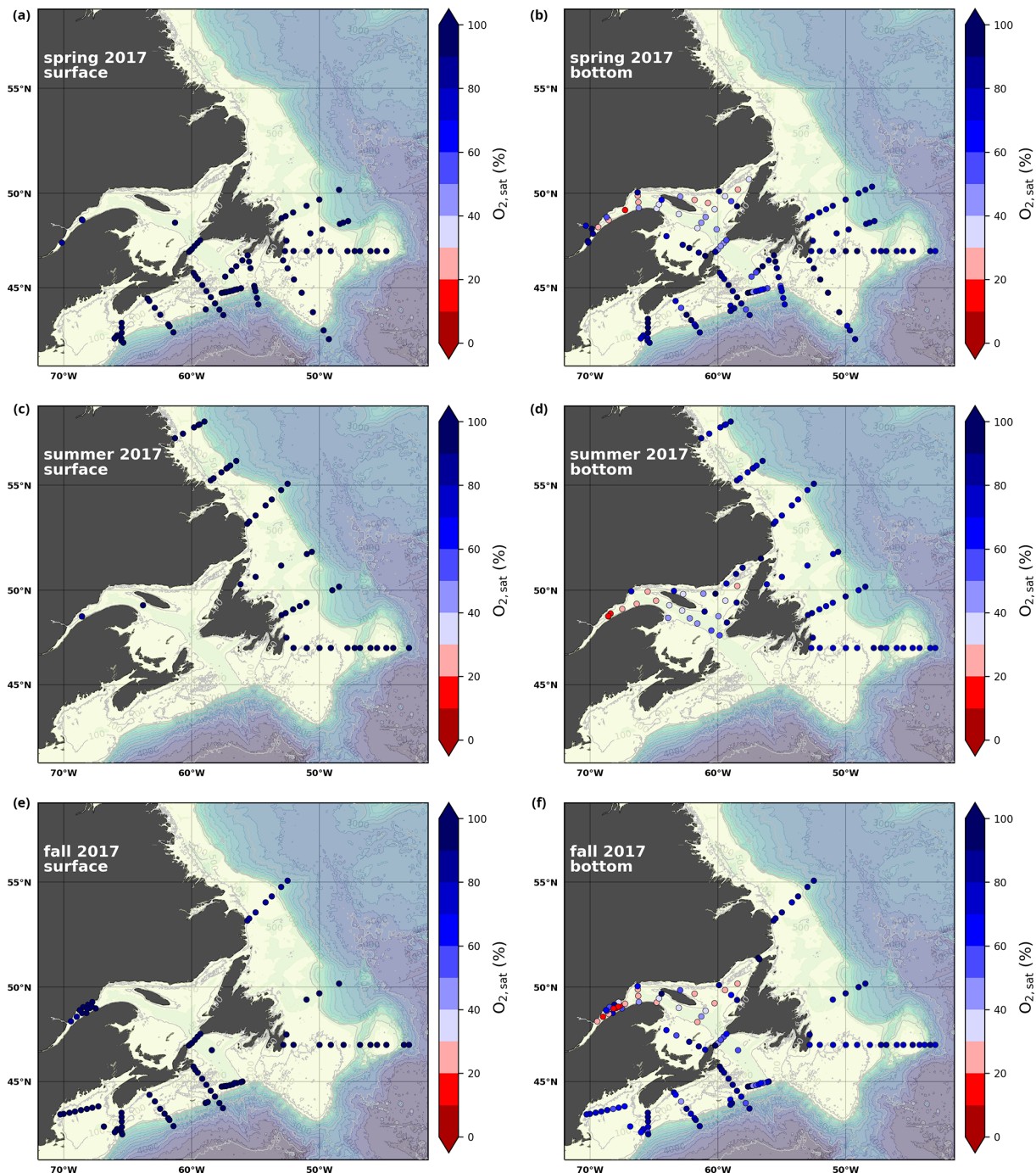

**Figure A3.** Same as in Fig. A1 but for $O_{2_{\text{sat}}}$.

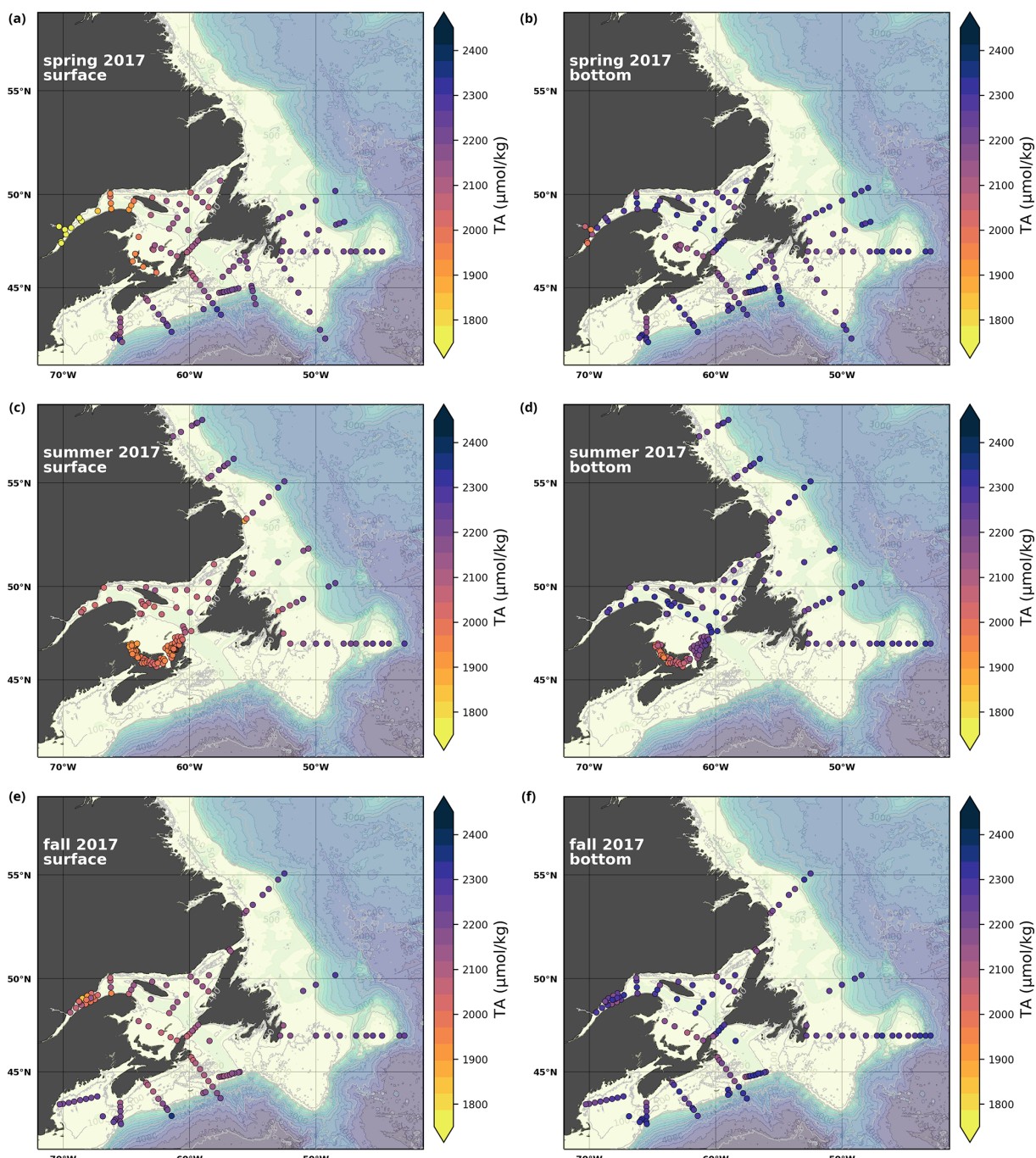

**Figure A4.** Same as in Fig. A1 but for TA.

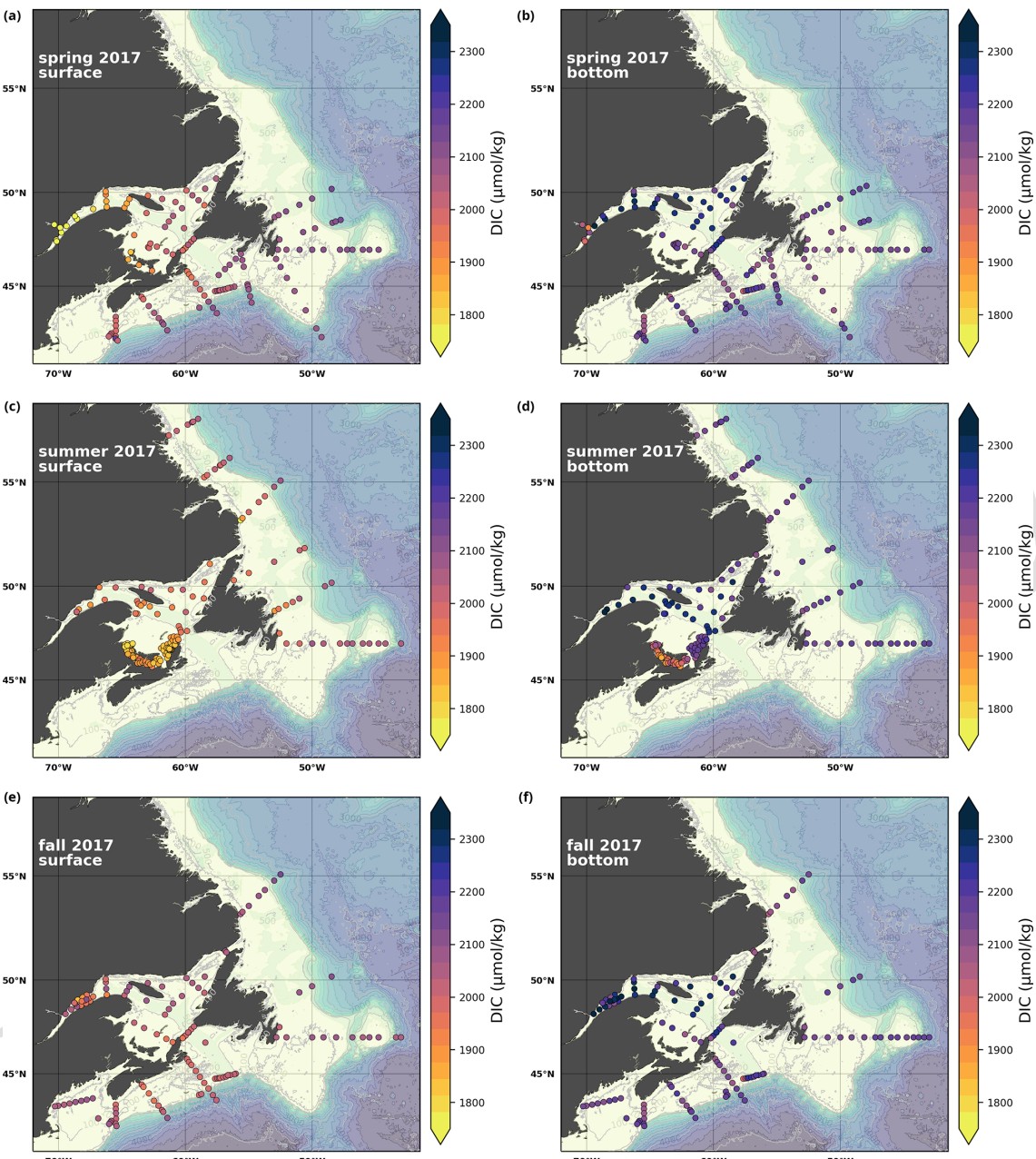

**Figure A5.** Same as in Fig. A1 but for TIC.

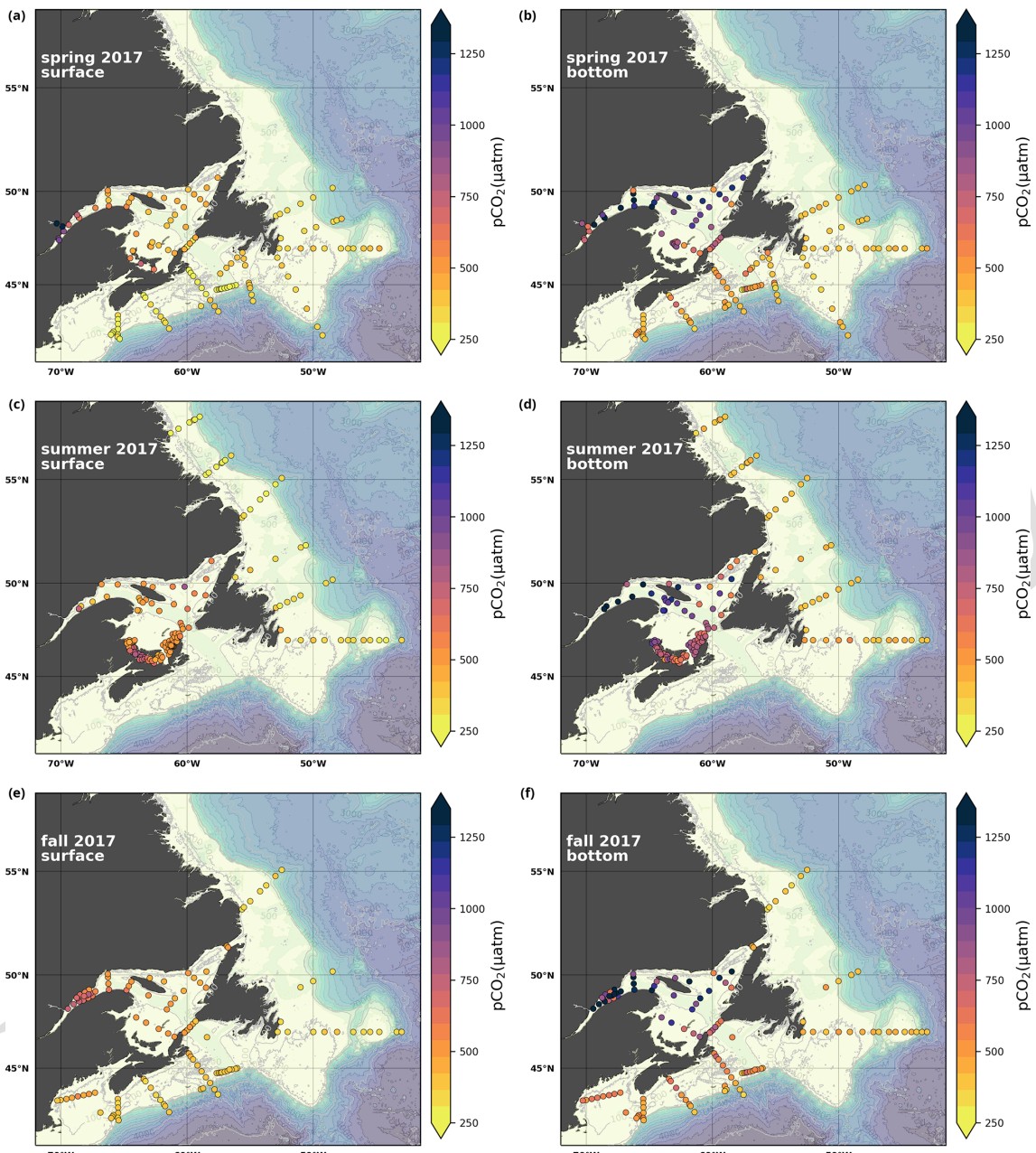

**Figure A6.** Same as in Fig. A1 but for $p$CO$_2$.

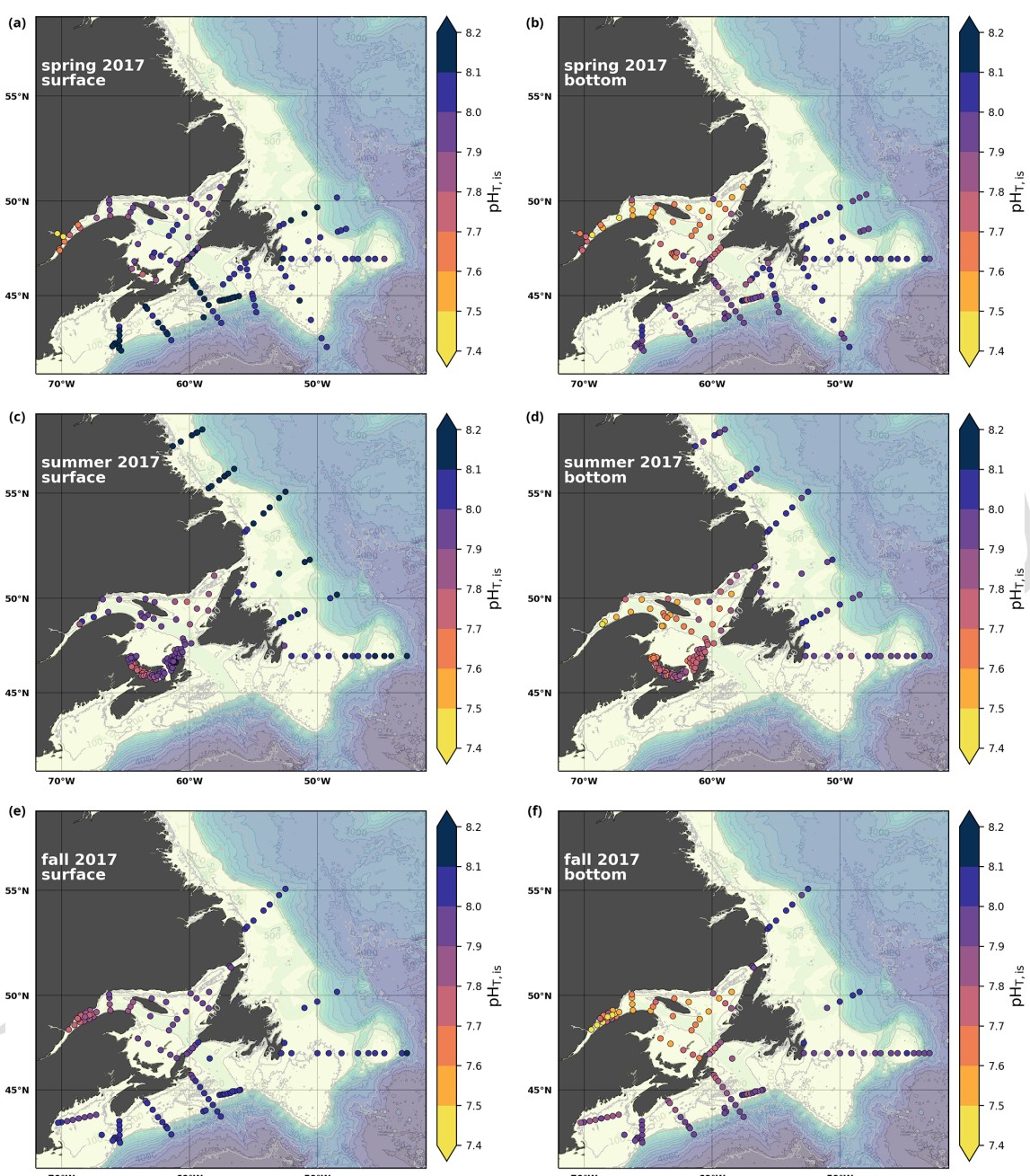

**Figure A7.** Same as in Fig. A1 but for pH.

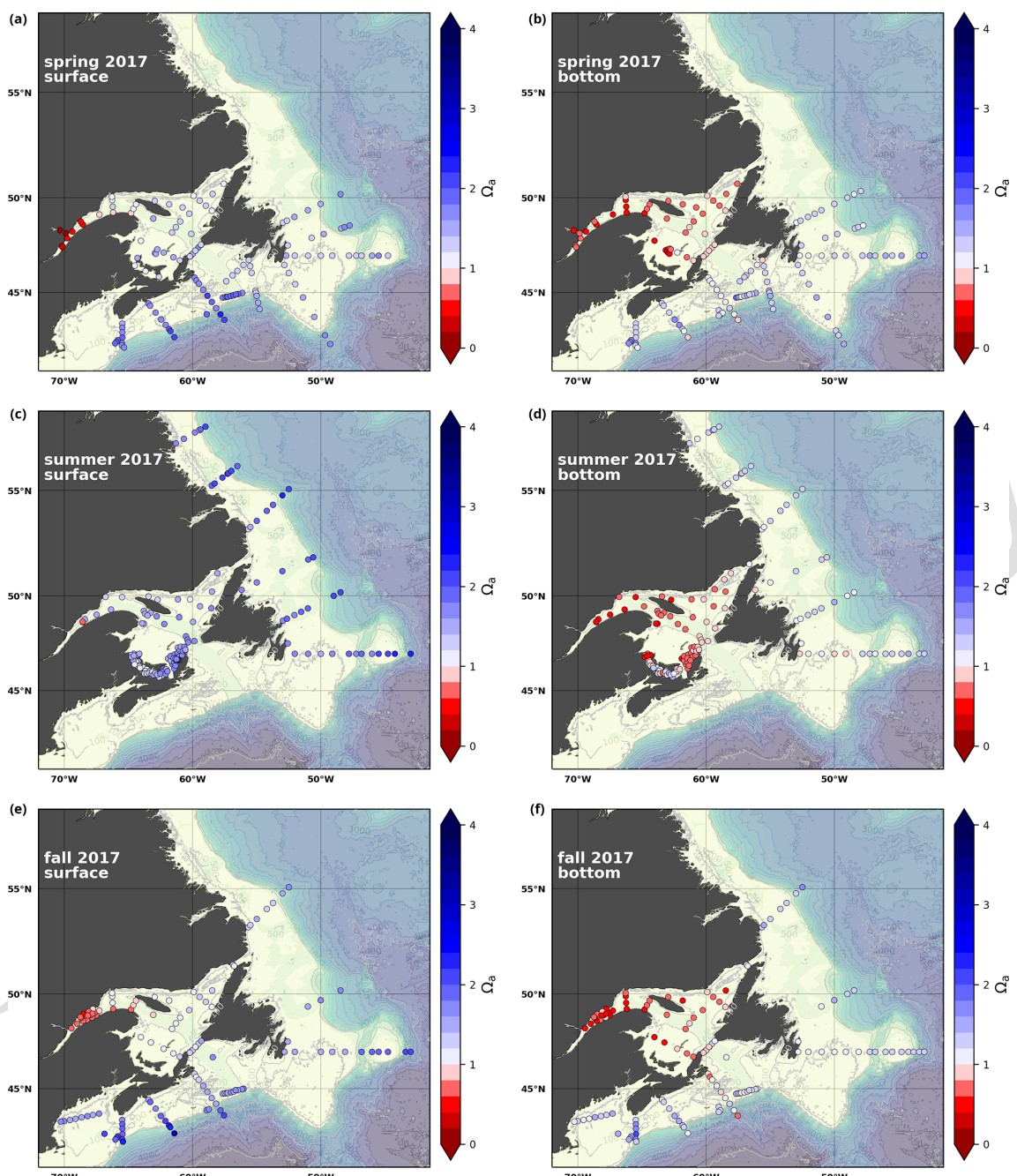

**Figure A8.** Same as in Fig. A1 but for $\Omega_{arg}$.

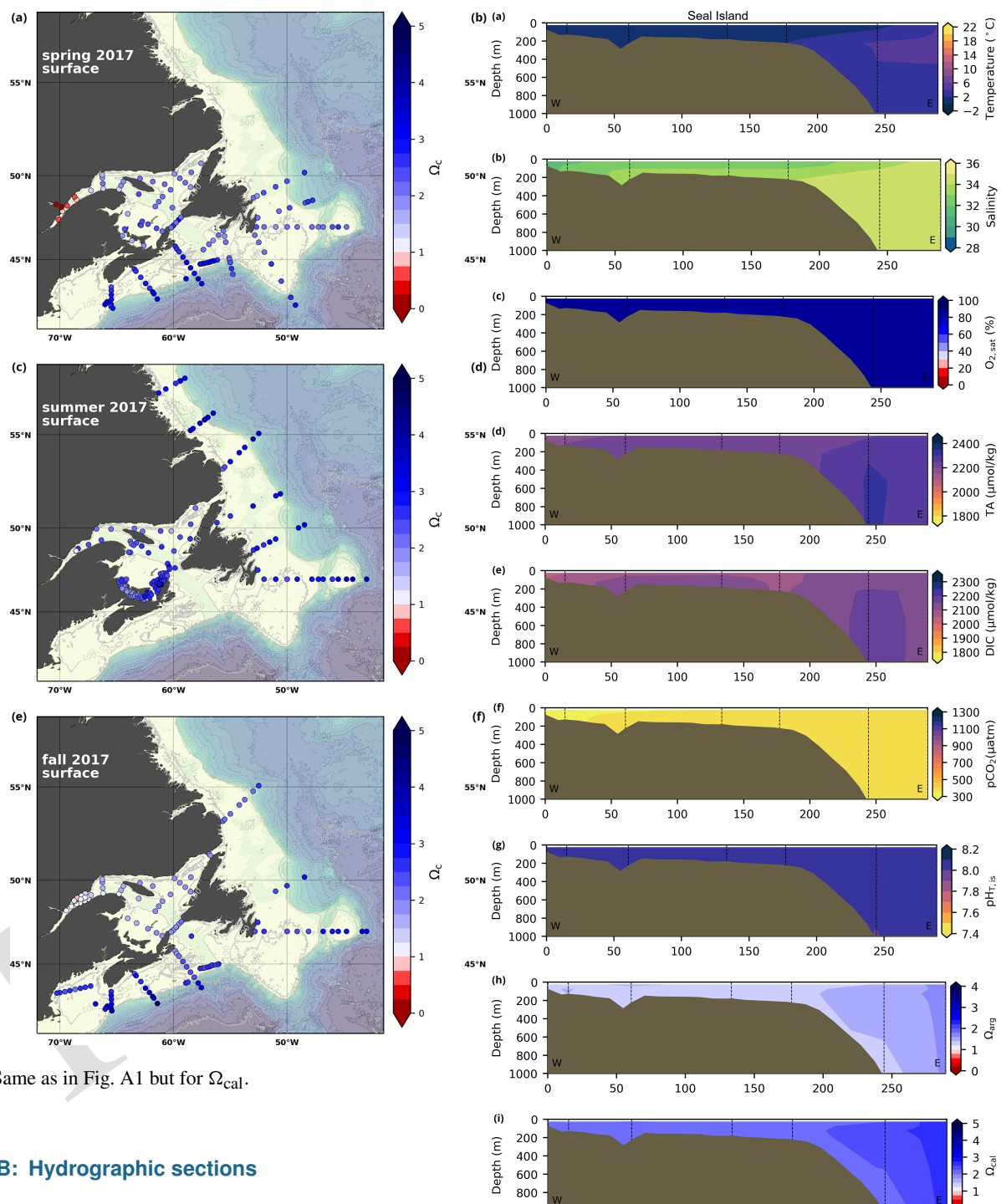

**Figure A9.** Same as in Fig. A1 but for $\Omega_{cal}$.

## Appendix B: Hydrographic sections

Figures B1 to B5 TS7 show contour plots of various physicochemical parameters collected as part of the AZMP (the same as in Appendix A) along selected hydrographic sections of the Atlantic Zone during the fall of 2017 (see Fig. 3 for location). These sections are Seal Island (SI), the Flemish Cap (FC), Halifax (HL), the Cabot Strait (CSL) and the Laurentian Channel (LC) and are presented in Figs. B1 to B5, respectively. Each figure shows, from top to bottom, the temperature, salinity, $O_{2_{sat}}$, TA, DIC, $pCO_2$, pH, $\Omega_{arg}$ and $\Omega_{cal}$, respectively.

**Figure B1.** Contour plots of various physicochemical parameters collected during the fall 2017 AZMP survey along the hydrographic section of Seal Island (NL region; see Fig. 3 for location). From top to bottom, these respective parameters are as follows: **(a)** temperature, **(b)** salinity, **(c)** $O_{2_{sat}}$, **(d)** TA, **(e)** DIC, **(f)** $pCO_2$, **(g)** pH, **(h)** $\Omega_{arg}$ and **(i)** $\Omega_{cal}$.

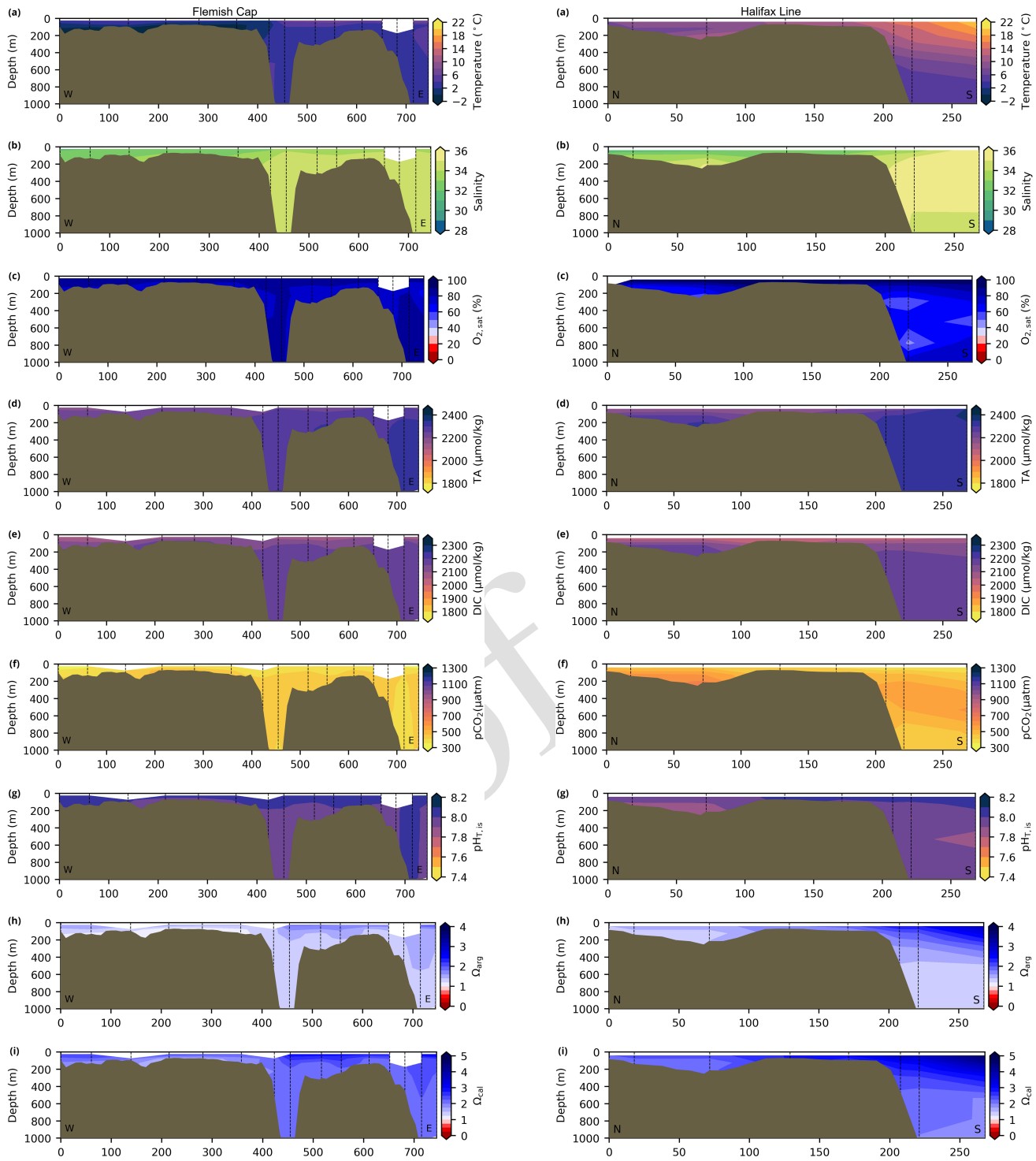

**Figure B2.** Same as in Fig. B1 but for the hydrographic section of Flemish Cap (NL region; see Fig. 3 for location).

**Figure B3.** Same as in Fig. B1 but for the hydrographic section of Halifax (MAR region; see Fig. 3 for location).

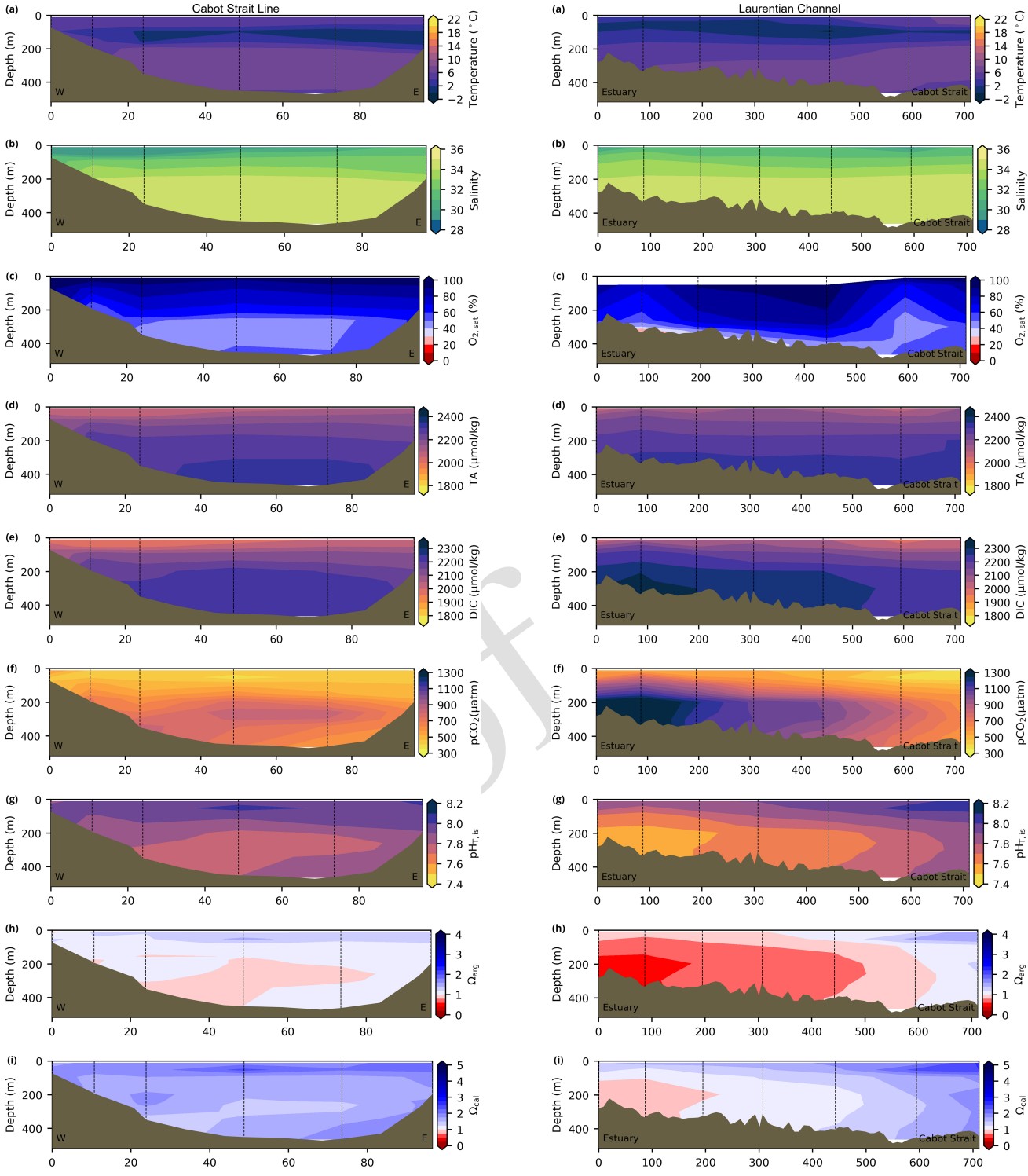

**Figure B4.** Same as in Fig. B1 but for the hydrographic section of Cabot Strait (GSL region; see Fig. 3 for location).

**Figure B5.** Same as in Fig. B1 but for the hydrographic section of the Laurentian Channel (GSL region; see Fig. 3 for location).

**Author contributions.** OG and FC led the study, the writing, and the data processing and archiving. GM; KAS, CEG and SP; and MS provided the quality-controlled data for Newfoundland and the Labrador Shelf, the Scotian Shelf, and the Gulf of St. Lawrence, respectively. PP, KAS and MS initiated the collection of carbonate parameters as part of the AZMP in 2014. All authors reviewed and commented on the manuscript.

**Competing interests.** The contact author has declared that none of the authors has any competing interests.

**Disclaimer.** Publisher's note: Copernicus Publications remains neutral with regard to jurisdictional claims in published maps and institutional affiliations.

**Acknowledgements.** This work is a contribution to the Atlantic Zone Monitoring Program (AZMP). The authors thank the numerous scientists, technicians, captains and crew members who have participated in the sampling and analysis effort since 2014. The authors would also like to thank Diane Lavoie and Jacqueline Dumas for their review and comments on an earlier version of this paper as well as the two anonymous reviewers for their insightful comments.

**Financial support.** This work was funded by the Fisheries and Oceans Canada Aquatic Climate Change Adaptation Services Program (ACCASP) under the "Delineation of Ocean Acidification and Calcium Carbonate Saturation State of the Atlantic Zone" (lead: Pierre Pepin, Stephen Snow and Kevin Anderson) and "Recent changes in the biogeochemistry of Northwest Atlantic water masses" (lead: Frédéric Cyr) projects. Frédéric Cyr also acknowledges support from the Natural Sciences and Engineering Research Council of Canada (NSERC) Advancing Climate Change Science in Canada program within the framework of the "Quantifying and predicting Canada's ocean carbon sink" (lead: Roberta Hamme) project. CE12

**Review statement.** This paper was edited by Giuseppe M. R. Manzella and reviewed by two anonymous referees.

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

## Remarks from the language copy-editor

CE1    Please confirm the slight change to the requested sentence.

CE2    Please confirm the addition.

CE3    Please note the change throughout. A slash would represent "or", whereas an en dash is grammatically correct (and is typeset differently from a minus sign). Please ensure that all instances have been changed correctly throughout the paper. Thank you.

CE4    "Scripps Institution of Oceanography" seems to be the official name on the website. Are you sure that the current change is correct? Please check.

CE5    Please confirm the addition.

CE6    Please confirm the in all instances.

CE7    Please confirm the change.

CE8    Please confirm the change.

CE9    Did you mean that "slope" should be lowercase here (extra comment: p.17, L.48: "slope" [see pdf] )? Please advise.

CE10    Please confirm the change in all instances.

CE11    Please confirm the change.

CE12    Please confirm the edits to this section.

## Remarks from the typesetter

TS1    Please confirm full name.

TS2    Thank you for your feedback. During file validation, the editor was asked to approve changes in this table, and they have been approved. However, changes that have to be approved by the editor have to be disclosed to the reader. This is only possible if post-review adjustments are conducted. Therefore, if this was the case, I would kindly ask you what was changed in the table and why.

TS3    Please confirm $r^2$.

TS4    Please confirm $r^2$.

TS5    Please confirm Figs. B1–B5 and their references in the text. Thank you.

TS6    Please confirm the caption.

TS7    Please confirm figure references.

TS8    Please confirm refrence list entry.

TS9    Please confirm reference list entry.

TS10    Please confirm reference list entry.

TS11    Please check the title.

TS12    Please confirm reference list entry.