# Peer review of "Spatiotemporal variability of pH and carbonate parameters on the Canadian Atlantic Continental Shelf between 2014 and 2022"

_Earth System Science Data, 2022_

## Author Comment (AC1)

**Reviewer 1 - Response from Authors**

*This data set is valuable and useful for future comprehensive analysis and modeling work along the Atlantic Coasts. The continuation of these efforts is essential to monitor ocean acidification and to support fisheries management.*

> Thank you!

*However, there are several minor comments that authors should pay attention to:*

*1). Equation (5), the expression of saturate state is wrong. And the symbol of the solubility product (Ksp or Ksp\*) is not consistent with the main text.*

> Our apologies, this is now corrected

*2). Line 115 – 120. The geographic names are not labeled in either Figure 1 or Figure 3, which leads to confusion. In addition, it is hard to distinguish the TCEN transect, CGSL, and LC transect from Figure 3, especially when the names are not well labeled. Also, several other geographic names are mentioned in the main text yet are not labeled in Figure 1 or 3.*

> Thanks for catching this, we will update the figures with the corresponding labels and make the transects clearer.

*3). Line 150 -155. The Rimouski Station and 26 are not found in Figure 1 or Figure 3*

> Thanks for catching this, we will update the figures.

*4). Line 190 -195. What equations are used to calculate pH from absorbance value? Are pH samples corrected for dye perturbation?*

> This will be clarified during the revision of the manuscript (the scientists responsible for these measurements are currently at sea).

> As far as we can tell, the 2014 pH across the entire Atlantic Zone was calculated following the equations with dye perturbation correction described in Dickson et al. (2007). Since 2019, pH measured in the Quebec region is not corrected for dye perturbation, but the m-cresol dye is visually inspected each day measurements are made to ensure its stability over time.

*5). Table 1. It is very useful to give a summary of the cruise information. However, I would strongly recommend adding one column to indicate what is measured carbonate parameters during the cruise because some syntheses only would like to use the measured data instead of extrapolated and calculated values.*

> Thanks for the suggestion, this will be added.

*5). The font size of the axis and color bar are too small in all the surface figures. Especially in Figures 6,7, 8. Also, the unit should be consistent as the main text (umol kg-1)*

This will be corrected

*6). Throughout the whole text including the figures, the p of pCO₂ should be in italics*

This is now corrected

*7). I would suggest Figure9 a and b as one figure and c and d as another since they are not focusing on the same issue.*

This will be done.

---

## Author Response (AR1)

Dear Dr. Manzella,

Thank you for your consideration of our manuscript essd-2022-460 entitled "Spatiotemporal variability of pH and carbonate parameters on the Canadian Atlantic Continental Shelf between 2014 and 2020" by Gibb et al.

We have carefully reviewed the comments from the two reviewers and are confident that we have satisfactorily answered all of them in our revision.

As part of the manuscript revision, we would also like to extend the dataset to the period 2014-2022 as new data are now available. In order to do so, we would also like to include two new co-authors who were responsible for the data in one region since the retirement of another co-author in 2020.

We provide below a point-by-point response to the Reviewers where our answers appear in blue below the copy-pasted comments of the Reviewers.

Yours sincerely,

Frédéric Cyr and Olivia Gibb on behalf of all co-authors

**Reviewer 1 - Response from Authors**

*This data set is valuable and useful for future comprehensive analysis and modeling work along the Atlantic Coasts. The continuation of these efforts is essential to monitor ocean acidification and to support fisheries management.*

> Thank you!

*However, there are several minor comments that authors should pay attention to:*

*1). Equation (5), the expression of saturate state is wrong. And the symbol of the solubility product (Ksp or Ksp\*) is not consistent with the main text.*

> Our apologies, this is now corrected

*2). Line 115 – 120. The geographic names are not labeled in either Figure 1 or Figure 3, which leads to confusion. In addition, it is hard to distinguish the TCEN transect, CGSL, and LC transect from Figure 3, especially when the names are not well labeled. Also, several other geographic names are mentioned in the main text yet are not labeled in Figure 1 or 3.*

> Thanks for catching this, the figures were updated with the corresponding labels. The transects are also clearer now with a different color scheme.

*3). Line 150 -155. The Rimouski Station and 26 are not found in Figure 1 or Figure 3*

> Thanks for catching this, the figures were updated.

*4). Line 190 -195. What equations are used to calculate pH from absorbance value? Are pH samples corrected for dye perturbation?*

> This is now clarified in the manuscript:

> *"In 2014 pH across the entire Atlantic Zone was calculated following the equations with dye perturbation correction described in Dickson et al. (2007). Since 2019, pH measured in the Quebec region is not corrected for dye perturbation, but the m-cresol dye is visually inspected each day measurements are made to ensure its stability over time."* See L.194

*5). Table 1. It is very useful to give a summary of the cruise information. However, I would strongly recommend adding one column to indicate what is measured carbonate parameters during the cruise because some syntheses only would like to use the measured data instead of extrapolated and calculated values.*

> Thanks for the suggestion, this is now added.

*5). The font size of the axis and color bar are too small in all the surface figures. Especially in Figures 6,7, 8. Also, the unit should be consistent as the main text (umol kg$_{-1}$)*

    Now corrected

*6). Throughout the whole text including the figures,  the p of pCO$_2$ should be in italics*

    Now corrected

*7). I would suggest Figure9 a and b as one figure and c and d as another since they are not focusing on the same issue.*

    Done.

**Reviewer 2 - Response from Authors**

*General Comments:*

*The paper provides an overview of carbonate chemistry parameters on the Canadian Atlantic Continental Shelf between 2014 and 2020. The authors have systematically measured at least two of the carbonate parameters in this region since 2014, allowing for the calculation of all carbonate chemistry. The paper presents valuable and important results in the context of the ongoing climate change and ocean acidification.*

    Thank you!

*However, there are some important points that must be addressed in this manuscript.*

*The main concern is with the figures' presentation and the general structure of this manuscript, particularly Figures 6, 7, and 8 (and discussion, see below). Despite the fact that the manuscript is well written, it is difficult to follow some of the authors' choices regarding the manuscript's structure. For example, why did the authors only show the maps of the surface and bottom of fall 2017 (Fig. 6 and 7), while the results for other seasons are in the annexes? In the same direction, it is not clear why only some parameters and seasons appear in Fig. 9. The authors should explain these choices more clearly in the main text. I suggest that the authors revise the acronyms used in the text, figure captions, and graphs, as they are not consistent.*

    The focus of this paper is to present a carbonate chemistry dataset on a large area
    of Atlantic Canada covering more than 1.7 million square km (about 3 times the area
    of France). This region is sampled by three administrative regions at different times
    of the year, with two regions sampling 3 times a year (spring, summer and fall), and

the other 2 times (e.g. spring and fall), but none exactly at the same time because the ship time is shared. This poses a challenge in terms of presentation of the data. Choices had to be made in order to provide the overview of the spatio-temporal variability of the different carbonate chemistry parameters while keeping a reasonable length in the text and in the number of figures. In the original submission, 24 Figures were provided (some of them included up to 9 sub-panels). The year 2017 was chosen as a "test case" because it is one of the years with the best coverage by the three regional groups (as stated in L.270). The fall (surface/bottom) is presented in Figures 6 to 8 for several variables, and all three seasons are presented in Appendix A (Figures A1 to A9; one figure per parameter). We now clearly state that this only represents a small portion of this dataset:

*"This section comprises of a brief description of the spatial and temporal variations of carbonate parameters of the Atlantic Zone. Except for 2015, the values and seasonal cycles among annual surveys are relatively comparable. For this reason, we chose the year 2017, one of the most complete year to date, both seasonally and spatially. In the following, descriptions made of the Atlantic Zone thus refers to maps and sections from 2017 (Figures 6-8, A1-A9)."* (L.325)

In a similar way, the dataset contains numerous variables that could be described. Choices were made to sometimes focus on one variable over multiple areas, while in some instances we chose one region to show multiple variables (Figures A10 to A14).

Finally, some interannual time series are presented at Station Rimouski and Station 27 where high frequency sampling is available.

We defer to the Editor for any further guidance on whether this choice is appropriate for this paper.

*I have two concerns about the carbonate chemistry data: (1) the TA-S relationship should not be used to derive the TA in sampling campaigns where TA was not measured, as the TA-S relationship shows an R2 that is far from the conservative mixing, particularly for data with salinity < 30; and (2) I strongly suggests that the authors perform a quality check of the carbonate chemistry calculations. In some oceanographic campaigns, the authors performed the analysis of the three-carbonate parameters (pH, DIC, TA). Therefore, the calculated values can be compared to the observed values for these three parameters. This is important because the water pCO2 was not directly measured*

See response to a similar comment below.

*Finally, the "Discussion and Conclusion" section is very short, and I don't know if this is a general recommendation of the ESSD. If not, I recommend that the authors go deeper into the discussion.*

> See next point

*Furthermore, many passages of the results section are discussion, so it is necessary a rearrangement and a review of these sections.*

> This paper aims to be a data paper accompanied with an overall description of the carbonate parameters of the Atlantic Zone. We did not pursue any particular research question (which would require an extended Discussion on a particular finding).
>
> That said, we propose to rename the current *Results* section "Results and Discussion" and rename the current *Discussion and Conclusion* section "Conclusion". We take advantage of this new structure to discuss each parameter as they are presented (see addition in cyan blue throughout this section).

*Specific comments*

*Abstract: The abstract is highly informative, but more in a qualitative way. I would like to see more quantitative information. Moreover, some statement about temporal trends (if any) along the studies period would be great.*

> Quantitative information is now provided in the abstract (see additions in cyan blue)

The Fig. 2 is great, but the labeling is very confused and not all acronyms are explained. Some acronyms appear in the figure, but not in the caption whereas some acronyms appear in the caption, but not in the figure. Some texts in the figure are difficult to read. Please revise.

> We revised the acronyms and the captions and the text.

The eq. 2 actually presents two equations of equilibrium. I recommend presenting:

$CO_2(aq) + H_2O = H^+ + HCO_3^-$

$HCO_3^- = H^+ + CO_3^{2-}$

> Thank you, now changed as suggested.

In the eq. 3 there is an error: it is not $2Ca^+$, it is $Ca^{2+}$. Please refer to Millero (2007).

> Thank you, now corrected.

There is another process, which is the dissolution and precipitation of CaCO3 that it is also very important in this context. The equation should appear in the text: Ca2+ + 2HCO3- = CaCO3 + CO2 + H2O.

> This process is already included in Equation 6 as well as in the accompanying text.

Line 73: "For example, the dilution of sea water decreases all carbonate parameters, except for pCO2." Avoid this type of generalization. Are you referring just to the Canadian Shelf? It should be explained. The freshwater endmember is highly variable and can show higher or lower values of DIC, TA, than open ocean waters. Overall, tropical rivers present lower TA concentrations than temperate/boreal rivers, thus the impact of thermodynamic processes during river-ocean mixing are largely variable (Cai et al., 2008; Cotovicz et al., 2020; Abril et al., 2021).

> Sorry for this generalization and thanks for the references provided, however these references discuss the very low salinities of riverine outflow. We simply removed this sentence from the manuscript.

Line 110. As I recommended before, please review all acronyms and be consistent in the text and in the figure captions.

> Thanks, now corrected.

Fig. 4. You should present the scale of pH (Total?).

> The scale is written on the y-axis as $pH_{is,T}$ which is pH in situ, total. This is now added to the Figure caption text.

Lines 220-230: You stated that "Since there are only a few surveys that include all three parameters (TA, pH and DIC), we have not produced the errors associated with the calculation of carbonate parameters because it would not be a representative calculation for the entire Atlantic Zone". I think the quality check is important for calculation purposes, particularly for water pCO2. I am not asking a complete error analysis, but some quality check is useful. As you used the TA-pH pair to calculate pCO2w, the presence of organic alkalinity (especially in nearshore, productive regions) can be significant and influence the pCO2 value. As you have some measurements including all three parameters (TA, pH and DIC), you can compare "calculated" x "observed" values.

> Sorry if this was unclear. We did perform the quality checks, but did not provide a full error analysis because only the GSL region collects all three parameters since 2019. In the revision of the manuscript, we however propose to add two more years of data collection, 2021 and 2022 (see the Letter to the Editor). This will increase the number of years where TA/DIC/pH are measured in the GSL (now 2019-2022) therefore providing more data for the analysis.

Regarding the Reviewer's point, we also added a new row (e.g., *d*) to Table 2 where the variability of pCO2 calculation using the different combinations TA/DIC/pH, TA/DIC and TA/pH for the data where all 3 parameters are measured. This is also discussed in the text near L.375 and 430.

Line 235: The TA can be estimated using the TA-salinity linear relationship only if the strength of the relationship is strong and almost conservative. The equation that you presented " TA = 41.26 x S + 870.05 (r2 = 0.93, p < 0.001)" is far from the conservative mixing, and probably reflects some biological processes of production and/or consumption of TA. The Fig. 5b shows that data for salinities < 30 are highly dispersed from linearity, and this should affect TA estimation. Therefore, I recommend to calculate TA using the TA-salinity only for salinities > 30, or to remove this data from the manuscript.

We understand that the reviewer is concerned with estimating TA using data with S<30. There are 76 samples from the dataset that did not have TA values. Unfortunately this was during the 2015 spring survey in the GSL, which produced 12 samples with salinity < 30. We have calculated the slopes of the curve with and without the lower salinity data and they are the same. We will therefore use only the S>30 data to calculate the TA for the 76 samples (see also Figure 5b). We would like to keep these data within the dataset and feel it is appropriate to do so since NOAA-OCADS accepts these data as long as it is clearly stated with corresponding metadata.

The molecular formula of some nutrients are wrong. Please correct.

Yes, we will correct this.

The structure of presentation of some Figures is not clear. For example, why did you present the fall 2017 maps in Figures 6 to 8, while for other seasons you present in Appendix A? Please refer also to the general comments.

See answer to General Comments

Lines 330-335: This is discussion, not results.

Now in Section "Results and Discussion"

Lines 343-345: This is also a discussion.

Now in Section "Results and Discussion"

Results for pH: As you have a high accuracy of pH measurements (0.005), I strongly recommend showing the results in two or three decimal places.

The results discuss the general regional and temporal patterns. I am not convinced that writing all of the pH values to 3 decimal places in the text is necessary to convey the message. However the values in the dataset are to 3 decimal places. There are some inconsistencies of significant figures in the dataset and these will be fixed according to measurement accuracy.

Lines 353-354: This is discussion, not results.

Now in Section "Results and Discussion"

Lines 356-357: "The surface water pH is relatively uniform throughout the Atlantic Zone in the fall (~7.8 to 8.1)." Not really.  The variability of 0.3 is not low.

The reviewer is correct, this was an oversight. We replaced by: "The surface water pH is relatively uniform for the NL shelf (range [8.06, 8.10]) while it varies between 7.88 and 8.08 on the Scotian shelf and between 7.74 and 7.98 for the GSL." (L.385)

Lines 362-364: This is discussion, not results.

Now in Section "Results and Discussion"

Lines 375-377: "The surface waters of the USLE are undersaturated ($\Omega$arg < 1) with respect to aragonite (and in some instances calcite) due to low TA and DIC". This is discussion, not results. In addition, this explanation is weak. Indeed, the saturation state decreases probably due to the decrease of TA compared to DIC. This can be investigate by plotting DIC/TA ratio against $\Omega$.

Yes this relationship is true when plotted (see Figure below). We updated the text to reflect this in the "Results and Discussion" section to say:

"The surface waters of the USLE are undersaturated ($\Omega$arg < 1) with respect to aragonite (and in some instances calcite) due to the observed decrease in TA and DIC, as well as the decrease in TA relative to DIC (as suggested by a scatter plot of DIC/TA vs $\Omega$ for the USLE; not shown)"

[Figure]

Lines 380-381: This is discussion, not results.

> Now in Section "Results and Discussion"

Discussion and Conclusion: This section is short, and I do not know if this is a characteristic of papers published in ESSD. If this is not the case, I recommend expanding this section. Furthermore, there are very few literature citations here. This should be reviewed. Finally, there are many sections of the "results" section that are, in fact, discussion. Therefore, the structure of the manuscript should be reviewed, and the discussion should be enhanced.

> See answer to General Comments about renaming "Results and Discussion" and "Conclusion" as separate sections.

---

## Author Response (AR2)

Dear Production Team,

This is regarding our manuscript essd-2022-460 entitled "Spatiotemporal variability of pH and carbonate parameters on the Canadian Atlantic Continental Shelf between 2014 and 2020" by Gibb et al.

This letter is to let you know that I provide here a slightly different version of the manuscript than the one accepted by the topical Editor. I already contacted the Editor by email on August 3rd 2023 to mention this.

The differences are very minor and are related to error calculations that appear in Table 2d (there is a reduction of the error following a bug found in our error calculation). The text was slightly modified to take into account these differences, but the main results of the study remains the same.

Thanks and Best Regards,
Frederic Cyr